# On the Sharp Input-Output Analysis of Nonlinear Systems under Adversarial Attacks

## Abstract

This paper is concerned with learning the input-output mapping of general nonlinear dynamical systems. While the existing literature focuses on Gaussian inputs and benign disturbances, we significantly broaden the scope of admissible control inputs and allow correlated, nonzero-mean, adversarial disturbances. With our reformulation as a linear combination of basis functions, we prove that the $\ell_2$-norm estimator overcomes the challenges posed by an adversary with access to the full information history, provided that the attack times are sparse, *i.e.*, the probability that the system is under adversarial attack at a given time is smaller than a certain threshold. We provide an estimation error bound that decays with the input memory length and prove its optimality by constructing a problem instance that suffers from the same bound under adversarial attacks. Our work provides a sharp input-output analysis for a generic nonlinear and partially observed system under significantly generalized assumptions compared to existing works.

## 1 Introduction

Dynamical systems describe how the state of a system evolves over time according to specific laws. Such systems are ubiquitous in scientific and engineering disciplines, including computer networks (Low et al., 2002), deep learning (Meunier et al., 2022), portfolio management (Grinold & Kahn, 2000), biology (Murray, 2007), and optimal control (Dorf & Bishop, 2011). In many practical settings, however, the underlying dynamics are too complex to be explicitly characterized, resulting in models with partially or entirely unknown parameters. Designing controllers or making predictions without first identifying these unknowns can lead to suboptimal or even unsafe outcomes. To address this challenge, the field of *system identification* focuses on identifying system dynamics from observed input-output data.

There has been extensive research in system identification under various structural and disturbance assumptions (Simchowitz et al., 2018; Faradonbeh et al., 2018; Simchowitz et al., 2019; Jedra & Proutiere, 2020; Sarkar et al., 2021; Oymak & Ozay, 2022; Bakshi et al., 2023; Yalcin et al., 2024; Zhang et al., 2025; Kim & Lavaei, 2025a;c) While these works provide strong theoretical guarantees and practical algorithms, the majority of them concentrate on linear systems. However, many real-world systems are inherently nonlinear (Grinold & Kahn, 2000; Low et al., 2002; Murray, 2007), which motivates us to develop identification methods that go beyond the linear setting.

We consider a generic partially observed nonlinear system

$$x_{t+1} = f(x_t, u_t, w_t), \qquad y_t = g(x_t, u_t), \quad t = 0, 1, \ldots, T-1, \tag{1}$$

where $x_t \in \mathbb{R}^n$ is the state, $u_t \in \mathcal{U} \subset \mathbb{R}^m$ is the control input, and $y_t \in \mathbb{R}^r$ is the observation at time $t$. The set $\mathcal{U}$ consists of admissible control inputs, and $T$ is the time horizon. The states evolve according to $f$, and (partial) observations from states are obtained from the states via $g$. Under *adversarially* chosen disturbances $w_t \in \mathbb{R}^d$, our goal is to identify the input-output mapping of the system (1) based on the collected data $\{u_t, y_t\}_{t=0}^{T-1}$. To be specific, given an input memory length $\tau > 0$, we study the mapping from the recent input sequence $(u_t, \ldots, u_{t-\tau})$ to the observation $y_t$.

In this paper, we approximate the input-output behavior of the system (1) using a finite-memory reformulation, which offers a tractable representation of the system under mild assumptions:

$$\text{(schematic)} \quad y_t = G^* \cdot \Phi(u_t, \ldots, u_{t-\tau}) + \textit{residual terms} + \textit{approximation error vector}, \tag{2}$$

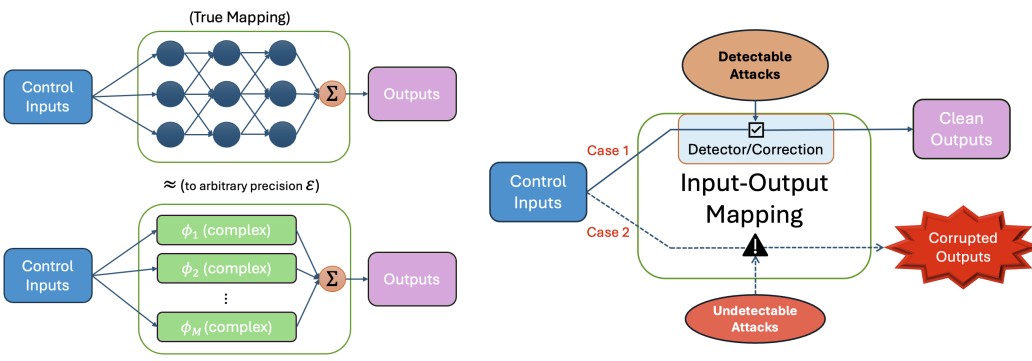

(a) Approximation of the true input-output mapping via a linearly parameterized mapping

(b) Impact of detectable attacks versus occasional undetectable attacks on outputs

Figure 1: Input-output analysis of linearly parameterized mappings under clean and corrupted outputs. (a) We reformulate the input-output behavior of nonlinear systems as a linearly parameterized nonlinear system, which can be approximated to arbitrary precision $\epsilon$, given a sufficiently expressive set of basis functions $\phi_1, \ldots, \phi_M$. (b) We assume that most attacks are detectable and produce clean outputs, whereas undetectable attacks occur infrequently but can have arbitrarily large magnitudes, producing completely corrupted outputs. Our goal is to identify a linearly parameterized input-output mapping from this partially corrupted output trajectory.

where $M$ is the number of basis functions, $\Phi : (\mathcal{U})^{\tau+1} \to \mathbb{R}^M$ is the stack of basis functions, $G^* \in \mathbb{R}^{r \times M}$ represents the matrix governing the true input-output mapping, $\tau > 0$ is input memory length, and the *residual terms* are functions of disturbances $w_{t-1}, \ldots, w_{t-\tau}$ and "far" past states $x_{t-\tau}$. Note that the far past states $x_{t-\tau}$ are exponentially small with the exponent $\tau$ under stability conditions. We will formalize this schematic form in Section 2.

In particular, given that the basis functions $\Phi(\cdot)$ are sufficiently expressive, it is guaranteed from the function approximation theory that a wide class of nonlinear mappings can be approximated to arbitrary precision, up to an *approximation error vector* of an arbitrarily small norm, using a finite set of appropriately chosen basis functions such as radial basis functions (RBF) (Chen et al., 1991), Volterra kernels (Boyd & Chua, 1985), and random feature models (Rahimi & Recht, 2007). Motivated by this, we focus on analyzing the corresponding linearly parameterized approximation of the input–output mapping (see Figure 1(a) for an overview and **Step 3** in Section 2 for the details).

Allowing for a small approximation error, we reduce the system identification task to estimating $G^*$. However, adversarial disturbances $w_t$, combined with the partial observability of the nonlinear system, introduce significant challenges to accurately recovering $G^*$. In cyberphysical systems, adversarial disturbances can be categorized as either detectable or undetectable attacks: the former are reliably detected and corrected by a well-designed detector and feedback controller (Fawzi et al., 2014; Shoukry & Tabuada, 2016; Pajic et al., 2017), whereas the latter—though injected occasionally—corrupt the outputs and hinder the identification of the mapping $G^*$ (see Figure 1(b)). Furthermore, it is natural to ask whether restrictions on admissible control inputs $u_t$ may also impede the identification task. To this end, we pose the following central question:

*When and how can we accurately estimate the true $G^*$ under*
*nonzero-mean, non-Gaussian inputs and correlated, nonzero-mean, adversarial disturbances?*

In this paper, we address the question posed above and summarize our contributions as follows:

1) Our work focuses on Lipschitz continuous nonlinear systems with partially observed outputs, non-Gaussian control inputs, and correlated, nonzero-mean, possibly adversarial disturbances. This setting significantly broadens the scope of existing literature, each of which assumes at least one of Gaussian control inputs, i.i.d. disturbances, or zero-mean disturbances. A detailed comparison of the problem setups is provided in Table 1.

2) We reformulate the problem as estimating $G^* \cdot \Phi(u_t, ..., u_{t-\tau})$, which represents a general form of modeling the system output as a linear combination of basis functions applied to a truncated input history of length $\tau$. When disturbances are fully adversarial at every time step, the matrix $G^*$

Table 1: Comparison of problem settings in existing literature with our work: N/A in Gaussian input means that they consider the system without inputs.

| | Dynamics | Available Outputs | Gaussian input | i.i.d. Disturbance | Zero-mean Disturbance | Identification Approach |
|---|---|---|---|---|---|---|
| Our Work | Nonlinear | Partial | No | No | No | Parametric |
| Sarkar et al. (2021) | Linear | Partial | Yes | Yes | Yes | Parametric |
| Oymak & Ozay (2022) | Linear | Partial | Yes | Yes | Yes | Parametric |
| Ziemann et al. (2022) | Nonlinear | Full | N/A | No | Yes | Nonparametric |
| Zhang et al. (2025) | Nonlinear | Full | N/A | No | Yes | Parametric |
| Kim & Lavaei (2025c) | Linear | Partial | Yes | No | No | Parametric |

becomes non-identifiable. Thus, within this framework, we characterize the class of problems for which the true $G^*$ can accurately be identified. In particular, we focus on the problems where the attack probability $p$ at each time (namely, the probability of $w_t$ being nonzero) is restricted to $p < \frac{1}{2\tau}$.

3) We establish that the estimation error of identifying $G^*$ using the $\ell_2$-norm estimator is $O(\rho^\tau)$, where $0 < \rho < 1$ is the contraction factor of the function $f$. Notably, we further provide a matching lower bound of $\Omega(\rho^\tau)$ on the estimation error, showing that the presented bound is indeed optimal.

**Related works.** We focus on identifying the input-output mapping of the system, since in many settings it suffices to capture how control actions influence observable outcomes (Abbeel et al., 2006; Deisenroth & Rasmussen, 2011). For instance, in model-based reinforcement learning (RL), the agent first learns an input–output model of the environment and subsequently uses it to make informed decisions (Moerland et al., 2023). To ensure tractability of our analysis, we adopt a parameterized system with a finite-memory approximation, which yields interpretable and computationally efficient models—particularly when the chosen function class closely aligns with the true system dynamics (Chen, 1995; Giannakis & Serpedin, 2001). The finite-memory approach is consistent with classical nonlinear system identification methods, such as Volterra series truncations (Boyd & Chua, 1985) and NARMAX models (Billings, 2013). Further details on related works are in Appendix A.

**Outline.** The paper is organized as follows. In Section 2, we formulate the problem and state the relevant assumptions. In Section 3, we prove that the $\ell_2$-norm estimator achieves the optimal estimation error and provides the analysis outline. In Section 4, we present numerical experiments to validate our main results. Finally, concluding remarks are provided in Section 5.

**Notation.** Let $\mathbb{R}^n$ denote the set of $n$-dimensional vectors and $\mathbb{R}^{n \times n}$ denote the set of $n \times n$ matrices. For a matrix $A$, $\|A\|_F$ denotes the Frobenius norm of the matrix. For a vector $x$, $\|x\|_2$ denotes the $\ell_2$-norm of the vector. For a set $S$, the $k$-fold Cartesian product $S \times S \times \cdots \times S$ (with $k$ factors) is denoted by $(S)^k$. For an event $E$, the indicator function $\mathbb{I}\{E\}$ equals 1 if $E$ occurs, and 0 otherwise. $\mathbb{P}(E)$ denotes the probability that the event occurs. We use $O(\cdot)$ for the big-$O$ notation and $\Omega(\cdot)$ for the big-$\Omega$ notation. Let $I_n$ denote the $n \times n$ identity matrix. The notation $\succeq$ denotes positive semidefiniteness. Let $N(\mu, \Sigma)$ denote the Gaussian distribution with mean $\mu$ and covariance $\Sigma$, and $\text{Unif}[a, b]^n$ denote the uniform distribution on the hypercube $[a, b]^n \subset \mathbb{R}^n$. Finally, let $\mathbb{E}$ denote the expectation operator.

## 2 PROBLEM FORMULATION

In (1), we study a nonlinear dynamical system $x_{t+1} = f(x_t, u_t, w_t)$ and $y_t = g(x_t, u_t)$, where the state equation is governed by the dynamics $f : \mathbb{R}^n \times \mathcal{U} \times \mathbb{R}^d \to \mathbb{R}^n$ and the observation equation is determined by the measurement model $g : \mathbb{R}^n \times \mathcal{U} \to \mathbb{R}^r$. We have the discretion to design control inputs $u_0, u_1, \ldots, u_{T-1}$ and we have access to a single observation trajectory consisting of partial observations $y_0, y_1, \ldots, y_{T-1}$. We assume the Lipschitz continuity of the measurement model $g$ and the contraction property for the dynamics $f$ to ensure system stability and prevent the explosion of the nonlinear system, which is common in control theory literature (Tsukamoto et al., 2021; Lin et al., 2023). The formal assumption on the dynamics is given below.

**Assumption 2.1** (Lipschitz Continuity). $g$ is Lipschitz continuous; *i.e.*, there exists $L > 0$ such that

$$\|g(x, u) - g(\tilde{x}, \tilde{u})\|_2 \leq L(\|x - \tilde{x}\|_2 + \|u - \tilde{u}\|_2) \tag{3}$$

for all $x, \tilde{x} \in \mathbb{R}^n$, $u, \tilde{u} \in \mathcal{U}$. Moreover, note that for $k \geq 1$, the $k$-fold composition of the dynamics $f$, denoted by $f^{(k)}$, maps $(x_{t-k}, u_{t-k}, \ldots, u_{t-1}, w_{t-k}, \ldots, w_{t-1})$ to $x_t$. We assume that $f^{(k)}$ is Lipschitz continuous in its oldest arguments $(x_{t-k}, u_{t-k}, w_{t-k})$ with constant $C\rho^k$ for some $C > 0$ and $0 < \rho < 1$, with later inputs and disturbances fixed. In other words, we have

$$\|f^{(k)}(x_{t-k}, u_{t-k}, w_{t-k}; \boldsymbol{u}, \boldsymbol{w}) - f^{(k)}(\tilde{x}_{t-k}, \tilde{u}_{t-k}, \tilde{w}_{t-k}; \boldsymbol{u}, \boldsymbol{w})\|_2$$
$$\leq C\rho^k(\|x_{t-k} - \tilde{x}_{t-k}\|_2 + \|u_{t-k} - \tilde{u}_{t-k}\|_2 + \|w_{t-k} - \tilde{w}_{t-k}\|_2), \quad (4)$$

for all $x_{t-k}, \tilde{x}_{t-k} \in \mathbb{R}^n, u_{t-k}, \tilde{u}_{t-k} \in \mathcal{U}, w_{t-k}, \tilde{w}_{t-k} \in \mathbb{R}^d$, with any $\boldsymbol{u} = (u_{t-k+1}, \ldots, u_{t-1}) \in (\mathcal{U})^{k-1}$ and $\boldsymbol{w} = (w_{t-k+1}, \ldots, w_{t-1}) \in (\mathbb{R}^d)^{k-1}$. We further make the standard assumption $f(0, 0, 0) = 0$.

**Remark 2.2.** The contraction property in Assumption 2.1 is analogous to Gelfand's formula in linear systems. For any matrix $A \in \mathbb{R}^{n \times n}$, the formula guarantees the existence of the absolute constant $c(n)$ (which only depends on the system order $n$) such that $\|A^k\|_2 \leq c(n) \cdot [\lambda_{\max}(A)]^k$ for all $k \geq 0$, where $\|\cdot\|_2$ denotes the spectral norm and $\lambda_{\max}(\cdot)$ denotes the spectral radius. We adopt this analogous setting in our nonlinear system by interpreting $\lambda_{\max}(A)$ as $\rho$, and assume that $f^{(k)}$ has a Lipschitz constant of $C\rho^k$.

In this work, we focus on input-output analysis and aim to identify the model governing the mapping from (truncated) control inputs $(u_t, \ldots, u_{t-\tau})$ to observation outputs $y_t$, where $\tau$ denotes the input memory length specified by the user to construct the mapping. As described in the introduction, we will reformulate the true mapping to a linearly parameterized input-output mapping with a finite-memory approximation. To this end, we outline the following four steps.

**Step 1** By recursively applying the system dynamics (1), the observation $y_t$ can be represented as

$$y_t = g(x_t, u_t) = g(f(x_{t-1}, u_{t-1}, w_{t-1}), u_t) = \cdots$$
$$= g(f(\cdots f(f(x_{t-\tau}, u_{t-\tau}, w_{t-\tau}), u_{t-\tau+1}, w_{t-\tau+1}), \ldots, u_{t-1}, w_{t-1}), u_t) \quad (5)$$

for all $t \geq \tau$, where the right-hand side considers the control inputs $u_t, u_{t-1}, \ldots, u_{t-\tau}$.

**Step 2** We next separate the effect of disturbances and the oldest state with the effect of inputs, which allows us to establish the input-output mapping. The result is summarized in the following lemma (see the proof in Appendix C).

**Lemma 2.3.** *Under Assumption 2.1, the equation* (5) *for each* $t \geq \tau$ *implies that there exist* $\boldsymbol{W}_t^{(\tau)}, \boldsymbol{x}_t^{(\tau)} \in \mathbb{R}^r$ *such that*

$$y_t = g(f(\cdots f(0, u_{t-\tau}, 0), \cdots, u_{t-1}, 0), u_t) + \boldsymbol{W}_t^{(\tau)} + \boldsymbol{x}_t^{(\tau)}, \quad (6)$$

*where* $\|\boldsymbol{W}_t^{(\tau)}\|_2 \leq CL\sum_{k=1}^{\tau} \rho^k \|w_{t-k}\|_2$ *and* $\|\boldsymbol{x}_t^{(\tau)}\|_2 \leq CL\rho^\tau \|x_{t-\tau}\|_2$.

**Step 3** For the equation (6), note that the term $g \circ f \circ \cdots \circ f$ is a function of $u_t, u_{t-1}, \ldots, u_{t-\tau}$. We convert this nonlinear function to a linear combination of basis functions taking a truncated number of control inputs, in which we establish a time-invariant system in the sense that the input memory length is fixed. We allow for a universal approximation tolerance $\bar{\epsilon} \geq 0$, such that $\|\epsilon_t(\cdot)\|_2 \leq \bar{\epsilon}$:

$$g(f(\cdots f(0, u_{t-\tau}, 0), \cdots, u_{t-1}, 0), u_t) = G^* \cdot [\phi_1(\boldsymbol{U}_t^{(\tau)}) \, \cdots \, \phi_M(\boldsymbol{U}_t^{(\tau)})]^T + \epsilon_t(\boldsymbol{U}_t^{(\tau)}), \quad (7)$$

where $\boldsymbol{U}_t^{(\tau)} = (u_t, \ldots, u_{t-\tau}) \in (\mathcal{U})^{\tau+1}$ is the stack of inputs from the time $t - \tau$ to $t$, the basis functions $\phi_i : (\mathbb{R}^m)^{\tau+1} \to \mathbb{R}$ are distinct nonlinear mappings for $i = 1, \ldots, M$, and the matrix $G^* \in \mathbb{R}^{r \times M}$ explains how the nonlinear transformation of the inputs is mapped to the observation outputs. The number of basis functions $M$ and how to design them can be chosen at the discretion of the user. We note that such a matrix $G^*$ is well-defined (though not necessarily unique) to represent the true input-output mapping within a small $\bar{\epsilon}$, given sufficiently expressive basis functions.

**Step 4** Let $\Phi(\boldsymbol{U}_t^{(\tau)}) := [\phi_1(\boldsymbol{U}_t^{(\tau)}) \, \cdots \, \phi_M(\boldsymbol{U}_t^{(\tau)})]^T$. Considering the relationships (6) and (7), we finally arrive at the equation

$$y_t = G^* \cdot \Phi(\boldsymbol{U}_t^{(\tau)}) + \boldsymbol{W}_t^{(\tau)} + \boldsymbol{x}_t^{(\tau)} + \epsilon_t(\boldsymbol{U}_t^{(\tau)}) \quad (8)$$

for all $t \geq \tau$. This provides an equivalent representation of $y_t$ via a linearly parameterized mapping (see Figure 1(a)), with the goal of accurately estimating the true matrix $G^*$ that governs the input-output mapping from the control inputs $\boldsymbol{U}_t^{(\tau)} = (u_t, \ldots, u_{t-\tau})$ to the observation $y_t$. $\qquad \square$

Function approximation theory ensures that the relationship (7) is valid since our function $g \circ f \circ \cdots \circ f$ is continuous. In particular, Assumption 2.1 (Lipschitz continuity) makes it natural to choose Lipschitz continuous basis functions such as polynomials or radial basis functions.

**Assumption 2.4** (Basis functions). Each basis function $\phi_i$ is designed to be $L_\phi$-Lipschitz, namely,

$$|\phi_i(\boldsymbol{U}_t^{(\tau)}) - \phi_i(\tilde{\boldsymbol{U}}_t^{(\tau)})| \leq L_\phi \|\boldsymbol{U}_t^{(\tau)} - \tilde{\boldsymbol{U}}_t^{(\tau)}\|_2, \quad \forall i = 1, \ldots, M, \tag{9}$$

for all $\boldsymbol{U}_t^{(\tau)}, \tilde{\boldsymbol{U}}_t^{(\tau)} \in (\mathcal{U})^{\tau+1}$. Also, each basis function with inputs should excite the system for the exploration in learning the system. In other words, there exists a universal constant $\lambda > 0$ such that

$$\mathbb{E}\left[\Phi(\boldsymbol{U}_t^{(\tau)})\Phi(\boldsymbol{U}_t^{(\tau)})^T\right] \succeq \lambda^2 I_M \tag{10}$$

holds for all $t \geq \tau$. We further assume that $\Phi(0) = 0$, meaning that zero input results in zero basis function values.

We also consider both inputs and disturbances on the system to be sub-Gaussian variables. For example, any bounded variables automatically satisfy this assumption. We use the definition given in Vershynin (2018) (see the definition, the $\psi_2$-norm, and properties of sub-Gaussian variables in Appendix B). Note that we do not require each input or disturbance to have a zero mean (see Definitions B.1 and B.4). The formal assumptions are given below.

**Assumption 2.5** (sub-Gaussian control inputs). We design our control input to be independent sub-Gaussian variables, meaning that $u_0, u_1, \ldots, u_{T-1}$ are independent of each other and there exists a finite $\sigma_u > 0$ such that $\|u_t\|_{\psi_2} \leq \sigma_u$ for all $t = 0, \ldots, T-1$.

**Assumption 2.6** (sub-Gaussian disturbances). Define a filtration

$$\mathcal{F}_t = \boldsymbol{\sigma}\{x_0, u_0, \ldots, u_t, w_0, \ldots, w_{t-1}\}.$$

Then, there exists $\sigma_w > 0$ such that $\|x_0\|_{\psi_2} \leq \sigma_w$ and $\|w_t\|_{\psi_2} \leq \sigma_w$ conditioned on $\mathcal{F}_t$ for all $t = 0, \ldots, T-1$ and $\mathcal{F}_t$.

**Remark 2.7.** While prior literature typically assumes zero-mean Gaussian inputs, we significantly relax these conditions by only requiring $u_t$ to be sub-Gaussian (see Assumption 2.5), and $\Phi(\boldsymbol{U}_t^{(\tau)})$ to be Lipschitz and excite the system (see Assumption 2.4). These assumptions characterize general conditions on control inputs for nonlinear system identification. In practice, control inputs are often of the form $K(y_t) + $ [excitation term] to improve performance (*e.g.*, minimize costs), where $K$ is a controller and $y_t$ is the observation at time $t$. Our characterization allows nonzero-mean $K(y_t)$ and non-Gaussian [excitation term], providing a secondary benefit for system identification.

If the disturbance $w_t$ is always adversarial with nonzero-mean, any estimator may be misled. For example, the adversary can always inject an attack $w_t$ that drives the next state $x_{t+1}$ to be irrelevant to the current state $x_t$, preventing any valid estimation method from extracting useful information (Kim & Lavaei, 2025a). Hence, we may need to restrict the time instances $t$ in which the adversary may be able to fully attack the system via $w_t$. We now formally present the additional restriction on our disturbances $w_t$, under which the input-output mapping in (8) is accurately estimated.

**Assumption 2.8** (Attack probability). $w_t$ is an adversarial attack at each time $t$ with probability $p < \frac{1}{2\tau}$ conditioned on $\mathcal{F}_t$; *i.e.*, there exists a sequence $(\xi_t)_{t\geq0}$ of independent Bernoulli($p$) variables, each independent of any $\mathcal{F}_t$, such that

$$\{\xi_t = 0\} \subseteq \{w_t = 0\}, \quad \forall t \geq 0. \tag{11}$$

Assumption 2.8 specifies that the system is not under attack (case 1 in Figure 1(b)) with probability at least $1 - p$, since $\xi_t = 0$ implies $w_t = 0$. At attack times (case 2 in Figure 1(b)), the adversary uses the information in the filtration $\mathcal{F}_t$ to generate disturbances $w_t$, which can therefore be correlated and possibly adversarial.

**Remark 2.9** (Choice of $\tau$). In Assumption 2.8, the attack probability depends on a user-defined constant $\tau$, which represents an input memory length. As discussed in the introduction, it is inevitable to consider a finite-memory approximation, and the permissible attack probability $\frac{1}{2\tau}$—which depends on the memory length $\tau$—will accordingly restrict the ability of the adversary. It is worth noting that the term $\boldsymbol{W}_t^{(\tau)}$ in (8) is identically zero if the system is not under attack for $\tau$ consecutive periods; *i.e.* $w_{t-1} = \cdots = w_{t-\tau} = 0$, which happens with probability at least $(1 - p)^\tau$. We have $(1 - p)^\tau > 0.5$ with the restriction on attack probability $p < \frac{1}{2\tau}$, which we will leverage to prove the useful results on the estimation error.

Given a time horizon $T$, we aim to learn the true system $G^*$ in (7)-(8), using the following $\ell_2$-norm estimator based on partial observations $\{y_t\}_{t=\tau}^{T-1}$ and control inputs $\{u_t\}_{t=0}^{T-1}$:

$$\hat{G}_T = \arg\min_G \sum_{t=\tau}^{T-1} \left\| y_t - G \cdot \Phi(\boldsymbol{U}_t^{(\tau)}) \right\|_2 \tag{12}$$

Under the stated assumptions, we will show in the next section that the $\ell_2$-norm estimator achieves the optimal estimation error $O(\rho^\tau)$, where $\rho$ is the contraction factor in Assumption 2.1 and $\tau$ is the input memory length.

## 3 MAIN THEOREMS AND ANALYSIS OUTLINE

In this section, we will state our main theorems on bounding the estimation error to identify $G^*$ and provide the analysis outline. Note that any $G^*$ satisfying (7) is regarded as a valid approximation to the true input-output mapping of the system (1).

### 3.1 MAIN THEOREM

Our main theorem holds under the stated assumptions, which incorporates non-Gaussian inputs and correlated, nonzero-mean, adversarial disturbances, with an attack probability $p$ no greater than $\frac{1}{2\tau}$.

**Theorem 3.1.** *Suppose that Assumptions 2.1, 2.4, 2.5, 2.6, and 2.8 hold. Consider $\nu := \frac{\sqrt{M\tau}L_\phi\sigma_u}{\lambda}$ and an approximation tolerance $\bar{\epsilon} \geq 0$. Let $G^*$ be any matrix that satisfies (7) with $\|\boldsymbol{\epsilon}_t(\boldsymbol{U}_t^{(\tau)})\|_2 \leq \bar{\epsilon}$ for all $t$. Also, let $\hat{G}_T$ denote a solution to the $\ell_2$-norm estimator given in (12). Given $\delta \in (0, 1]$, when*

$$T = \Omega\left( \frac{\tau\nu^8}{(2(1-p)^\tau - 1)^2} \left[ rM \log\left( \frac{\tau\nu}{2(1-p)^\tau - 1} \right) + \log\left( \frac{1}{\delta} \right) \right] \right), \tag{13}$$

*we have*

$$\|G^* - \hat{G}_T\|_F = O\left( \left( \frac{\rho^\tau L}{\lambda} \cdot \frac{\sigma_u + \sigma_w}{1 - \rho} + \frac{\bar{\epsilon}}{\lambda} \right) \cdot \frac{\nu^3}{2(1-p)^\tau - 1} \right) \tag{14}$$

*with probability at least $1 - \delta$.*

**Remark 3.2.** Our main theorem states that after the time given in (13) and with a sufficiently small $\bar{\epsilon}$, the estimation error of $O(\rho^\tau)$ is achieved, considering that additional polynomial terms in $\tau$ are dominated by the exponential decay in $\tau$. However, notice that the estimation error does not decay as the time $T$ increases, and thus cannot converge to zero. While this error bound decreases as $\tau$ grows, the memory length $\tau$ will be chosen as a finite number at the user's discretion, and thus the bound should be treated as a positive constant. This suggests that the user may want to choose a sufficiently long $\tau$ to obtain a smaller estimation error. However, increasing $\tau$ has three drawbacks: First, it restricts the attack probability as stated in Remark 2.9. Second, the required time (13) implies that it takes longer to arrive at the desired estimation estimation bound. Third, the basis function $\Phi$ may become significantly complex to incorporate longer history, and naturally the optimization problem needs far more computations. Thus, even though the estimation error may decrease with increasing $\tau$, the aforementioned demerits create an inherent trade-off in selecting an appropriate value for $\tau$.

### 3.2 ANALYSIS OUTLINE

We now provide the outline of proof analysis. The proof details can be found in Appendix D.

#### 3.2.1 ANALYSIS WITHOUT PAST STATE AND APPROXIMATION EFFECT

Our proof technique starts from a special case where the term $\boldsymbol{x}_t^{(\tau)}$ and $\boldsymbol{\epsilon}_t$ in the equation (8) are zero. This auxiliary setting will later be generalized to the case where they can take nonzero values. In the following theorem, we establish a sufficient condition for the true matrix $G^*$ to be the unique solution to the $\ell_2$-norm minimization problem (12).

**Theorem 3.3.** *Suppose that $\boldsymbol{x}_t^{(\tau)} = 0$ and $\boldsymbol{\epsilon}_t = 0$ for all $t$. Then, $G^*$ is the unique solution to the $\ell_2$-norm estimator* (12) *if*

$$\sum_{t=\tau}^{T-1} \left\| Z\Phi(\boldsymbol{U}_t^{(\tau)}) \right\|_2 \cdot \mathbb{I}\{\boldsymbol{W}_t^{(\tau)} = 0\} - \sum_{t=\tau}^{T-1} \left\| Z\Phi(\boldsymbol{U}_t^{(\tau)}) \right\|_2 \cdot \mathbb{I}\{\boldsymbol{W}_t^{(\tau)} \neq 0\} > 0, \qquad (15)$$

*holds for all $Z \in \mathbb{R}^{r \times M}$ such that $\|Z\|_F = 1$.*

Theorem 3.3 implies that if the left-hand side given in equation (15) is positive for all $Z \in \mathbb{R}^{r \times M}$ such that $\|Z\|_F = 1$, we will actually be able to exactly recover the true matrix $G^*$ with the $\ell_2$-norm estimator. In particular, thanks to Lemma 2.3 and (11), we have

$$\mathbb{P}(\boldsymbol{W}_t^{(\tau)} = 0) \geq \mathbb{P}(w_{t-1} = 0, \ldots, w_{t-\tau} = 0) \geq \mathbb{P}(\xi_{t-1} = 0, \ldots, \xi_{t-\tau} = 0) = (1-p)^\tau > 0.5.$$

Then, for a fixed $Z$, the sub-Gaussianity of control inputs $u_t$, Lipschitzness of $\Phi(\cdot)$, and the excitation condition (10) ensures that the left-hand side of (15) will be sufficiently positive after a finite time.

We now analyze how the term in (15) changes when evaluated at two different points $Z, \tilde{Z} \in \mathbb{R}^{r \times M}$. We show that the difference is indeed small when the points are close. Thus, if one can select a sufficient number of points for which the term in (15) is simultaneously positive with high probability, then their surrounding neighborhoods will also yield positive values. This implies that the term in (15) is universally positive for all points in $\mathbb{R}^{r \times M}$ with unit Frobenius norm. To quantify how many such points are needed, we invoke a well-known covering number argument (Vershynin, 2018). $\qquad \square$

### 3.2.2 Beyond the Zero Past State and Approximation Effect

In general, $\boldsymbol{x}_t^{(\tau)}$ in (8) is nonzero since $\|x_{t-\tau}\|_2 \neq 0$ (see Lemma 2.3). Moreover, we may face a nonzero approximation error vector $\boldsymbol{\epsilon}_t$, whose magnitude depends on the expressiveness of the chosen basis functions. Thus, we need to extend the previous analysis in Section 3.2.1 to general cases. From the optimality of $\hat{G}_T$ for the $\ell_2$-norm estimator (12) and the input-output mapping (8), we have

$$\sum_{t=\tau}^{T-1} \|(G^* - \hat{G}_T)\Phi(\boldsymbol{U}_t^{(\tau)}) + \boldsymbol{W}_t^{(\tau)} + \boldsymbol{x}_t^{(\tau)} + \boldsymbol{\epsilon}_t\|_2 \leq \sum_{t=\tau}^{T-1} \|\boldsymbol{W}_t^{(\tau)} + \boldsymbol{x}_t^{(\tau)} + \boldsymbol{\epsilon}_t\|_2, \qquad (16)$$

where the right-hand side is the result of substituting $G^*$ into $G$ in (12). Using the triangle inequality, we can arrive at

$$\sum_{t=\tau}^{T-1} \|(G^* - \hat{G}_T)\Phi(\boldsymbol{U}_t^{(\tau)}) + \boldsymbol{W}_t^{(\tau)}\|_2 - \|\boldsymbol{W}_t^{(\tau)}\|_2 \leq 2 \sum_{t=\tau}^{T-1} \left( \|\boldsymbol{x}_t^{(\tau)}\|_2 + \|\boldsymbol{\epsilon}_t\|_2 \right) \qquad (17)$$

where the left-hand side turns out to be the perturbation of $\ell_2$-norm estimator without the effect of $\boldsymbol{x}_t^{(\tau)}$ and $\boldsymbol{\epsilon}_t$. This can be lower-bounded by using the *positive constant lower bound* $\Omega(T)$ of the term in (15) constructed in the previous Section 3.2.1. The right-hand side is also upper-bounded by $O(T)$ since disturbances and control inputs are sub-Gaussian variables (see Lemma D.9). Accordingly, we can bound the estimation error $\|G^* - \hat{G}_T\|_F$ using (17) to obtain the results in Theorem 3.1. $\qquad \square$

**Remark 3.4.** We note that any $\ell_\alpha$-norm estimator with $\alpha \geq 1$ can ensure the left-hand side of (15) remains universally positive even though $\ell_2$-norm is replaced by other norms. However, the resulting estimation error bound is weaker than that of the $\ell_2$-norm estimator. We analyze two different cases:

Case 1 — $1 \leq \alpha < 2$: In this regime, the final estimation error bound (14) suffers from an additional multiplicative factor, at most $\sqrt{r}$. This arises from the inequality $\|\boldsymbol{x}_t^{(\tau)}\|_1 + \|\boldsymbol{\epsilon}_t\|_1 \leq \sqrt{r}(\|\boldsymbol{x}_t^{(\tau)}\|_2 + \|\boldsymbol{\epsilon}_t\|_2)$, which will appear in (17) and ultimately worsens the estimation error bound.

Case 2 — $2 < \alpha \leq \infty$: Our analysis hinges on $\mathbb{E}\big[\|Z\Phi(\boldsymbol{U}_t^{(\tau)})\|_2^2\big] \geq \lambda^2$ (see (38)) for the $\ell_2$-norm estimator. Using $\alpha > 2$ also introduces an additional factor of $\sqrt{r}$, since the term of interest in the worst case is $\|Z\Phi(\boldsymbol{U}_t^{(\tau)})\|_\infty \geq \frac{1}{\sqrt{r}}\|Z\Phi(\boldsymbol{U}_t^{(\tau)})\|_2 \geq \frac{\lambda}{\sqrt{r}}$, which negatively affects the estimation error bound.

Consequently, the $\ell_2$-norm estimator yields a tighter estimation bound $O\big(\frac{\rho^\tau L}{\lambda}\big)$ than those based on other norms since it does not depend on the observation dimension $r$. This will further be supported by the lower bound presented in the next subsection, which is also independent of $r$.

### 3.3 LOWER BOUND

In this section, we claim that there is no estimator that can improve the constant bound in Theorem 3.1 in the worst case.

**Theorem 3.5.** *Given $\delta \in (0, 1]$, suppose that adversarial attacks $w_t$ are designed by an adversary to satisfy $\sigma_w = \left(\frac{1}{\rho}\right)^{\Omega(\tau \log(T/\delta))}$, where $0 < \rho < 1$ is the contraction factor of $f$, and $\tau$ is the input memory length. Then, there exists a problem instance satisfying Assumptions 2.1, 2.4, 2.5, 2.6, and 2.8 that suffers from $\Omega\left(\frac{\rho^\tau L}{\lambda}\right)$ estimation error with probability at least $1 - \delta$ for any estimator.*

*Proof Sketch.* Consider the case of an approximation tolerance $\bar{\epsilon} = 0$. Under the attack probability $O(1/\tau)$ in Assumption 2.8, the maximum consecutive attack-free length is bounded by $O(\tau \log(T/\delta))$ with probability at least $1 - \delta$. Then, the adversarial attacks enable the property $x_t \geq 1$ for all $t$ with high probability. This implies that $y_t$ in (5) can be written in two different functions $h_1 \neq h_2$ such that

$$y_t = h_1(x_{t-\tau}, u_{t-\tau}, \ldots, u_t, w_{t-\tau}, \ldots, w_{t-1}) = h_2(x_{t-\tau}, u_{t-\tau}, \ldots, u_t, w_{t-\tau}, \ldots, w_{t-1}) \quad (18)$$

for all $x_{t-\tau} \geq 1$, which implies that $h_1$ and $h_2$ are not distinguishable under adversarial attacks. However, the corresponding input-output mappings (see (6)) will be

$$h_1(0, u_{t-\tau}, \ldots, u_t, 0, \ldots, 0) \quad \text{and} \quad h_2(0, u_{t-\tau}, \ldots, u_t, 0, \ldots, 0), \quad (19)$$

where $x_{t-\tau}$ and the disturbances are set to 0. Choose the functions $h_1$ and $h_2$ to have different function values for (19), while satisfying the equation (18) for all $x_{t-\tau} \geq 1$. As a result, any estimator may recover either one of the mappings $h_1$ or $h_2$ arbitrarily, given the same observation trajectory $y_0, y_1, \ldots, y_{T-1}$. In particular, the two expressions in (19) can differ by $\Omega(\rho^\tau L)$, leading to an estimation error $\Omega(\frac{\rho^\tau L}{\lambda})$. The proof details can be found in Appendix E. □

**Remark 3.6.** We have established the lower bound $\Omega(\frac{\rho^\tau L}{\lambda})$, which implies that the estimation error is bounded below by a positive constant for any estimator due to adversarial attacks. While this matches the upper bound (14) up to the same order, the assumption on the sub-Gaussian norm of the attacks depends on $T$. If this norm is required to be uniformly bounded over all $T \geq 0$, it remains unclear whether the gap between the upper and lower bounds can be further tightened. Meanwhile, the proof in Appendix E relies on specially designed nonlinear basis functions to achieve the desired lower bound. It remains an intriguing open question whether there exists a linear system instance for which the upper and lower bounds match under the constraint that all basis functions are linear in control inputs.

## 4 NUMERICAL EXPERIMENTS

In this section, we provide the numerical experiments that show the effectiveness of the $\ell_2$-norm estimator and illustrate how the results align with our theoretical findings. To this end, we consider the following dynamics with the states $x_t \in \mathbb{R}^{100}$, the inputs $u_t \in \mathbb{R}^5$, the disturbances $w_t \in \mathbb{R}^{100}$, and the observations $y_t \in \mathbb{R}^{10}$ for $t = 0, \ldots, T - 1$:

$$x_{t+1} = f(x_t, u_t, w_t) = \sigma(Ax_t + Bu_t + w_t), \quad y_t = g(x_t, u_t) = Cx_t + Du_t, \quad (20)$$

where $A \in \mathbb{R}^{100 \times 100}$, $B \in \mathbb{R}^{100 \times 5}$, $C \in \mathbb{R}^{10 \times 100}$, $D \in \mathbb{R}^{10 \times 5}$ are randomly selected matrices and the function $\sigma(x) = \tanh(x)$ is 1-Lipschitz and is applied elementwise to each coordinate. Each entry of $A, B, C$ and $D$ is randomly selected from $\text{Unif}[-1, 1]$ and $A$ is normalized subsequently to have a spectral radius less than one (see Assumption 2.1 and Remark 2.2). As a result, the $\tau$-fold composition of $f$ will have the form of a feedforward neural net, where $\sigma(\cdot)$ works as an activation function. For the system (20), the relevant input-output mapping in (6) can be written as

$$\sigma(C\sigma(A\sigma(\cdots \sigma(A\sigma(Bu_{t-\tau}) + Bu_{t-\tau+1}) \cdots) + Bu_{t-1}) + Du_t). \quad (21)$$

We first reformulate the true input-output mapping as a linear combination of basis functions $G^* \cdot \Phi(\cdot)$ (see (8)). Our chosen basis functions are polynomial kernels up to degree 3, using randomly sampled tuples $(u_t, \ldots, u_{t-\tau})$ whose entries are drawn from $\text{Unif}[-15, 15]$. We then use kernel regression to estimate the true matrix $G^*$. The number of kernels used as basis functions is set to $M = 25$.

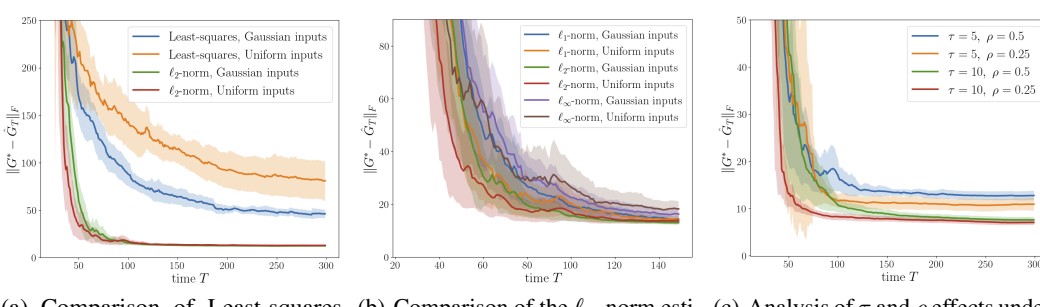

(a) Comparison of Least-squares and $\ell_2$-norm estimator

(b) Comparison of the $\ell_\alpha$-norm estimators ($\alpha = 1, 2, \infty$)

(c) Analysis of $\tau$ and $\rho$ effects under non-Gaussian inputs

Figure 2: Estimation error of the input-output mapping (21) under adversarial attacks.

*Experiment 1.* The first experiment compares the $\ell_2$-norm estimator with a standard least-squares estimator. The attack probability is set to $\frac{1}{2\tau+1}$, with sub-Gaussian attack $w_t$ designed to have a covariance $25I_{100}$ and a mean vector whose entries are either $300$ or $1000$ depending on the sign of the corresponding coordinate of $x_t$. Figure 2(a) shows that least-squares is vulnerable to attacks and fails to recover the system, while the $\ell_2$-norm estimator closely identifies the system after finite time. Results are provided under both Gaussian inputs $N(0, 100I_5)$ and non-Gaussian (nonzero-mean) inputs $\text{Unif}[-8, 10]^5$. Our theory only requires Assumptions 2.4 and 2.5 on the inputs, which is supported by our results showing that the $\ell_2$-norm estimator converges to the same stable region for the Gaussian inputs even when using the nonzero-mean non-Gaussian inputs (see Table 1).

*Experiment 2.* Under the same experimental settings, we now present experiments comparing the $\ell_\alpha$-norm estimators, where $\alpha = 1, 2, \infty$. As discussed in Remark 3.4, all norm estimators are expected to recover the true matrix $G^*$ to some extent, but only the $\ell_2$-norm estimator theoretically achieves the optimal error $O(\rho^\tau)$ that matches the lower bound $\Omega(\rho^\tau)$ (see Theorems 3.1 and 3.5). Figure 2(b) indeed verifies that the $\ell_2$-norm estimator outperforms the other norm estimators, although the empirical differences are relatively small.

*Experiment 3.* We finally provide experiments under different hyperparameters: the contraction factor $\rho$ and the input memory length $\tau$, using the $\ell_2$-norm estimator. Figure 2(c) demonstrates how the estimation error evolves over time under non-Gaussian inputs considered in *Experiment 1*. The figure illustrates that a larger $\rho$ results in a higher estimation error, while a larger $\tau$ leads to a smaller eventual estimation error. These two observations align precisely with an estimation error of $O(\rho^\tau)$—increasing with $\rho$ and decreasing with $\tau$. It is worth noting that this estimation error does not decay over time in the figures, which strongly supports the constant lower bound $\Omega(\rho^\tau)$. In Appendix F, we provide experimental details along with additional results for the case where an unbounded function is designed as the activation function $\sigma$.

In Appendix G, we present real-world experiments to identify the input-output mapping of the nonlinear swing dynamics in a power grid with $n$ generators, where an adversary can occasionally apply arbitrary power injections.

## 5 CONCLUSION

In this paper, we study the identification of the input-output mappings of nonlinear dynamical systems, where control inputs are not necessarily Gaussian and the disturbances are potentially adversarial. We formulate a time-invariant input-output mapping using a linear combination of basis functions taking the input history, where we decouple the control inputs and disturbances. We propose a problem class that accurately identifies the input-output mapping, characterized by a restriction on the attack probability. We then prove that the estimation error using $\ell_2$-norm estimator amounts to $O(\rho^\tau)$ under the presence of adversarial attacks and show that this bound is optimal by providing a matching lower bound $\Omega(\rho^\tau)$. Future directions include extending our analysis to a nonparametric approach under the same assumptions, where the estimator inherently involves an infinite-dimensional problem such as optimization over a function class.

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

## A  DETAILS ON RELATED WORKS

*Fully and Partially Observed Systems.*  In system identification, based on the degree of state observability, systems are often categorized as fully observed and partially observed systems. In fully observed systems, all states are measured, thus the outputs are identical to the states. In such systems, numerous methods have been proposed to recover the underlying system, e.g., least-squares methods (Simchowitz et al., 2018; Faradonbeh et al., 2018; Jedra & Proutiere, 2020), $\ell_2$-norm estimator (Yalcin et al., 2024; Zhang et al., 2025), and $\ell_1$-norm estimator (Kim & Lavaei, 2025a). However, in many real-world applications—such as robotics (Lauri et al., 2023), healthcare (Alagoz, 2014), and safety-critical systems (Bensoussan, 1992)—not all states are observable, giving rise to the partially observed system setting. In this case, system identification becomes substantially more challenging. A growing body of research has addressed this challenge: for instance, Sarkar et al. (2021); Oymak & Ozay (2022) used least-squares and Bakshi et al. (2023) used a method-of-moments estimator to identify the system, all under the assumption that disturbances are independent and follow Gaussian or sub-Gaussian distribution with zero-mean. Simchowitz et al. (2019) extended the least-squares method to setups where the disturbances can be selected by an oblivious adversary with access to past history (but not full information history). However, little research has been conducted in the partially observed systems when the disturbances are adversarially selected based on full information history. Only recently, Kim & Lavaei (2025b;c) investigated system identification using the $\ell_2$-norm or $\ell_1$-norm estimator and allowed adversarial disturbances to leverage full information history, while restricting the number of attack times. Our work adopts a similar assumption in Assumption 2.8.

*Nonparametric and Parametric approaches.*  Nonlinear system identification approaches can generally be classified into two broad categories: nonparametric and parametric methods. Nonparametric approaches operate over infinite-dimensional function spaces, often leveraging techniques such as kernel methods and deep learning to model complex system dynamics (Greblicki & Pawlak, 2008; Ziemann et al., 2022). These methods are highly flexible, making them well-suited for capturing behaviors without strong structural assumptions. However, they often come with significant computational overhead and reduced interpretability. In contrast, our approach is based on parametric methods that approximate the system using a finite set of basis functions, typically chosen based on prior knowledge or structural insights. This approach yields models that are more interpretable, computationally efficient, and easier to analyze—especially when the chosen function class aligns well with the true system dynamics (Chen, 1995; Giannakis & Serpedin, 2001). Moreover, the parametric framework facilitates model selection and regularization, enabling effective control over model complexity and reducing the risk of overfitting through techniques such as cross-validation or penalization.

*Finite-memory approximation.*  For a tractable identification of input-output mappings, we adopt a finite-memory approximation strategy with length $\tau$, consistent with classical system identification techniques such as Volterra series truncations (Boyd & Chua, 1985) and NARMAX models (Billings, 2013). These methods are grounded in the assumption that the dynamics of a nonlinear system can effectively be represented using a fixed window of past inputs.

*Input-output mapping.*  In many cases, it suffices to focus on the input-output relationship—how control actions affect observable outcomes—rather than attempting to recover the full latent state dynamics (Abbeel et al., 2006; Deisenroth & Rasmussen, 2011). Similarly, to identify the input-output mapping, we design the basis functions to depend solely on the control inputs, thereby decoupling the control inputs and disturbances. This separation has proven effective and is widely adopted in various settings. For example, in model-based reinforcement learning (RL), it is common to alternate between system identification and control policy design, where the agent first learns an input-output model of the environment and then uses it to make informed decisions (Moerland et al., 2023). This simplification is particularly valuable in high-stakes applications like autonomous driving, where control inputs such as throttle and steering are mapped to observations such as heading direction, position, and velocity (Paden et al., 2016).

## B  PRELIMINARIES ON SUB-GAUSSIAN VARIABLES

In this work, we consider both inputs and attacks on the system to be sub-Gaussian variables in which the tail event rarely occurs. We use the definition given in Vershynin (2018).

**Definition B.1** (sub-Gaussian scalar variables). A random variable $w \in \mathbb{R}$ is called sub-Gaussian if there exists $c > 0$ such that

$$\mathbb{E}\Big[\exp\Big(\frac{w^2}{c^2}\Big)\Big] \leq 2. \tag{22}$$

Its sub-Gaussian norm is denoted by $\|w\|_{\psi_2}$ and defined as

$$\|w\|_{\psi_2} = \inf\Big\{c > 0 : \mathbb{E}\Big[\exp\Big(\frac{w^2}{c^2}\Big)\Big] \leq 2\Big\}. \tag{23}$$

Note that the $\psi_2$-norm satisfies properties of norms: positive definiteness, homogeneity, and triangle inequality. We have the following properties for a sub-Gaussian variable $w$:

$$\mathbb{E}\big[|w|^k\big] \leq (C_1\sqrt{k})^k \quad \forall k = 1, 2, \ldots, \tag{24a}$$

$$\mathbb{P}(|w| \geq s) \leq 2\exp(-s^2/C_2^2), \quad \forall s \geq 0, \tag{24b}$$

$$\mathbb{E}[\exp(\theta w)] \leq \exp(\theta^2 C_3^2), \quad \forall \theta \in \mathbb{R} \quad \text{if } \mathbb{E}[w] = 0, \tag{24c}$$

where $C_1, C_2, C_3$, and $\|w\|_{\psi_2}$ are positive absolute constants that differ from each other by at most an absolute constant factor. For example, there exist $K, \tilde{K} > 0$ such that $c_1, c_2, c_3 \leq K\|w\|_{\psi_2}$ and $\|w\|_{\psi_2} \leq \tilde{K}\max\{c_1, c_2, c_3\}$. Note that the property (24b) is also called Hoeffding's inequality, which can be split into two inequalities if $\mathbb{E}[w] = 0$:

$$\mathbb{P}(w \geq s) \leq \exp(-s^2/C_2^2), \quad \forall s \geq 0, \tag{25a}$$

$$\mathbb{P}(w \leq -s) \leq \exp(-s^2/C_2^2), \quad \forall s \geq 0. \tag{25b}$$

We introduce the following useful lemmas to analyze the sum of independent noncentral sub-Gaussians (Vershynin, 2018).

**Lemma B.2** (Centering lemma). *If $w$ is a sub-Gaussian variable satisfying* (22), *then $w - \mathbb{E}[w]$ is also a sub-Gaussian variable with*

$$\|w - \mathbb{E}[w]\|_{\psi_2} = O(\|w\|_{\psi_2}). \tag{26}$$

**Lemma B.3** (Sum of mean-zero independent sub-Gaussians). *Let $w_1, \ldots, w_N$ be independent, mean zero, sub-Gaussian random variables. Then, $\sum_{i=1}^{N} w_i$ is also sub-Gaussian and its sub-Gaussian norm is $O\big((\sum_{i=1}^{N} \|w_i\|_{\psi_2}^2)^{1/2}\big)$.*

To provide the analysis of high-dimensional systems, we introduce the notion of sub-Gaussian vectors below.

**Definition B.4** (sub-Gaussian vector variables). A random vector $w \in \mathbb{R}^d$ is called sub-Gaussian if for every $x \in \mathbb{R}^d$, $w^T x$ is a sub-Gaussian variable. Its norm is defined as

$$\|w\|_{\psi_2} = \sup_{\|x\|_2 \leq 1, x \in \mathbb{R}^d} \|w^T x\|_{\psi_2}. \tag{27}$$

For example, if $w$ is a sub-Gaussian vector with a norm $\gamma$, then the sub-Gaussian norm of $\|w\|_2$ is also $\gamma$, considering that $w^T \frac{w}{\|w\|_2} = \|w\|_2$.

Throughout the paper, we will assume that the inputs and attacks injected into the system are indeed sub-Gaussian vectors. For example, the $m$-dimensional Gaussian variables and the $r$-dimensional bounded attacks are indeed sub-Gaussian vectors.

We finally define a notion of subexponential, which is essentially a squared sub-Gaussian.

**Lemma B.5.** *$w$ is sub-Gaussian if and only if $w^2$ is subexponential, and it holds that*

$$\|w\|_{\psi_2}^2 = \|w^2\|_{\psi_1},$$

*where the $\psi_1$-norm is defined as*

$$\|w^2\|_{\psi_1} = \inf\Big\{c > 0 : \mathbb{E}\Big[\exp\Big(\frac{w^2}{c}\Big)\Big] \leq 2\Big\}. \tag{28}$$

## C    PROOF OF LEMMA 2.3

We first provide the proof of Lemma 2.3, which states that the observation equation can be separated into the input term consisting of $(u_t, \ldots, u_{t-\tau})$, the attack term $\boldsymbol{W}_t^{(\tau)}$, and the oldest state term $\boldsymbol{x}_t^{(\tau)}$.

*Proof.* We iteratively apply (4) to the equation (5) for $k = \tau, \tau - 1, \ldots, 1$. For $k = \tau$, one can write

$$\|y_t - g(f(\cdots f(f(0, u_{t-\tau}, 0), u_{t-\tau+1}, w_{t-\tau+1}), \cdots, u_{t-1}, w_{t-1}), u_t)\|_2 \leq CL\rho^\tau(\|x_{t-\tau}\|_2 + \|w_{t-\tau}\|_2),$$

since the composition of $g$ and $f^{(k)}$ yields a Lipschitz function with Lipschitz constant $L \cdot C\rho^k$. In turn, for $k = \tau - 1$, we have

$$\|g(f(\cdots f(f(0, u_{t-\tau}, 0), u_{t-\tau+1}, w_{t-\tau+1}), \cdots, u_{t-1}, w_{t-1}), u_t)$$
$$- g(f(\cdots f(f(0, u_{t-\tau}, 0), u_{t-\tau+1}, 0), \cdots, u_{t-1}, w_{t-1}), u_t)\|_2 \leq CL\rho^{\tau-1}\|w_{t-\tau+1}\|_2.$$

Similarly, one can derive the similar inequalities for $k = \tau - 2, \ldots, 2$ and finally arrive at $k = 1$:

$$\|g(f(\cdots f(f(0, u_{t-\tau}, 0), u_{t-\tau+1}, 0), \cdots, u_{t-1}, w_{t-1}), u_t)$$
$$- g(f(\cdots f(f(0, u_{t-\tau}, 0), u_{t-\tau+1}, 0), \cdots, u_{t-1}, 0), u_t)\|_2 \leq CL\rho\|w_{t-1}\|_2.$$

Note that $g(f(\cdots f(f(0, u_{t-\tau}, 0), u_{t-\tau+1}, 0), \cdots, u_{t-1}, 0), u_t)$ is the auxiliary observation, where the attacks and the oldest state are assumed to be zero. Let $\bar{y}_t$ denote the difference between the true observation and the auxiliary observation:

$$\bar{y}_t := y_t - g(f(\cdots f(f(0, u_{t-\tau}, 0), u_{t-\tau+1}, 0), \cdots, u_{t-1}, 0), u_t).$$

Then, summing up all the inequalities for $k = \tau, \ldots, 1$ and applying the triangle inequality to the left-hand side implies that

$$\|\bar{y}_t\|_2 \leq CL\rho^\tau \|x_{t-\tau}\|_2 + CL \sum_{k=1}^{\tau} \rho^k \|w_{t-k}\|_2. \tag{29}$$

Now, we define the following random variables:

$$\boldsymbol{W}_t^{(\tau)} := \frac{\sum_{k=1}^{\tau} \rho^k \|w_{t-k}\|_2}{\rho^\tau \|x_{t-\tau}\|_2 + \sum_{k=1}^{\tau} \rho^k \|w_{t-k}\|_2} \bar{y}_t, \quad \boldsymbol{x}_t^{(\tau)} := \frac{\rho^\tau \|x_{t-\tau}\|_2}{\rho^\tau \|x_{t-\tau}\|_2 + \sum_{k=1}^{\tau} \rho^k \|w_{t-k}\|_2} \bar{y}_t. \tag{30}$$

Notice that $\bar{y}_t = \boldsymbol{W}_t^{(\tau)} + \boldsymbol{x}_t^{(\tau)}$. This implies that $\|\bar{y}_t\|_2 = \|\boldsymbol{W}_t^{(\tau)} + \boldsymbol{x}_t^{(\tau)}\|_2 \leq \|\boldsymbol{W}_t^{(\tau)}\|_2 + \|\boldsymbol{x}_t^{(\tau)}\|_2$, where each term $\|\boldsymbol{W}_t^{(\tau)}\|_2$ and $\|\boldsymbol{x}_t^{(\tau)}\|_2$ is bounded by the quantity in the lemma due to (29) and (30). □

## D    PROOF OF THEOREM 3.1

For convenience, we define $\mathbb{I}_\pm(\cdot)$ as the indicator function that equals 1 if the event occurs and $-1$ otherwise.

**Theorem D.1** (Restatement of Theorem 3.3). *Suppose that $\boldsymbol{x}_t^{(\tau)} = 0$ and $\boldsymbol{\epsilon}_t = 0$ for all t. Then, $G^*$ is the unique solution to the $\ell_2$-norm estimator* (12) *if*

$$\sum_{t=\tau}^{T-1} \mathbb{I}_\pm\{\boldsymbol{W}_t^{(\tau)} = 0\} \cdot \big\|Z\Phi(\boldsymbol{U}_t^{(\tau)})\big\|_2 > 0 \tag{31}$$

*holds for all $Z \in \mathbb{R}^{r \times M}$ such that $\|Z\|_F = 1$.*

*Proof.* Since $\boldsymbol{x}_t^{(\tau)} = 0$ and $\boldsymbol{\epsilon}_t = 0$, an equivalent condition for $G^*$ to be the unique solution of the convex optimization problem (12) is the existence of some $\bar{\Delta} > 0$ such that

$$\sum_{t=\tau}^{T-1} \|\boldsymbol{W}_t^{(\tau)}\|_2 < \sum_{t=\tau}^{T-1} \|\Delta \cdot \Phi(\boldsymbol{U}_t^{(\tau)}) + \boldsymbol{W}_t^{(\tau)}\|_2, \quad \forall \Delta \in \mathbb{R}^{r \times M} : 0 < \|\Delta\|_F \leq \bar{\Delta}, \tag{32}$$

since a strict local minimum in convex problems implies the unique global minimum. Observe that we have

$$
\sum_{t=\tau}^{T-1} \|\Delta \cdot \Phi(\boldsymbol{U}_t^{(\tau)}) + \boldsymbol{W}_t^{(\tau)}\|_2 - \sum_{t=\tau}^{T-1} \|\boldsymbol{W}_t^{(\tau)}\|_2
$$

$$
= \sum_{t=\tau}^{T-1} \mathbb{I}\{\boldsymbol{W}_t^{(\tau)} = 0\} \cdot \|\Delta \cdot \Phi(\boldsymbol{U}_t^{(\tau)})\|_2 + \mathbb{I}\{\boldsymbol{W}_t^{(\tau)} \neq 0\} \cdot \left( \|\Delta \cdot \Phi(\boldsymbol{U}_t^{(\tau)}) + \boldsymbol{W}_t^{(\tau)}\|_2 - \|\boldsymbol{W}_t^{(\tau)}\|_2 \right)
$$

$$
\geq \sum_{t=\tau}^{T-1} \mathbb{I}\{\boldsymbol{W}_t^{(\tau)} = 0\} \cdot \|\Delta \cdot \Phi(\boldsymbol{U}_t^{(\tau)})\|_2 + \mathbb{I}\{\boldsymbol{W}_t^{(\tau)} \neq 0\} \cdot \left( -\|\Delta \cdot \Phi(\boldsymbol{U}_t^{(\tau)})\|_2 \right)
$$

$$
= \sum_{t=\tau}^{T-1} \mathbb{I}_{\pm}\{\boldsymbol{W}_t^{(\tau)} = 0\} \cdot \left\| \Delta \cdot \Phi(\boldsymbol{U}_t^{(\tau)}) \right\|_2. \tag{33}
$$

Thus, a sufficient condition for (32) is to satisfy

$$
\sum_{t=\tau}^{T-1} \mathbb{I}_{\pm}\{\boldsymbol{W}_t^{(\tau)} = 0\} \cdot \left\| \Delta \cdot \Phi(\boldsymbol{U}_t^{(\tau)}) \right\|_2 > 0, \quad \forall \Delta \in \mathbb{R}^{r \times M} : 0 < \|\Delta\|_F \leq \bar{\Delta}. \tag{34}
$$

For each $\Delta$, dividing both sides of (34) by $\|\Delta\|_F > 0$ leads to the set of inequalities in (31). $\qquad\square$

We will now analyze the sub-Gaussian norm of $\|\boldsymbol{U}_t^{(\tau)}\|_2$.

**Lemma D.2.** *Under Assumption 2.5, we have* $\left\| \|\boldsymbol{U}_t^{(\tau)}\|_2 \right\|_{\psi_2} \leq \sqrt{\tau + 1} \cdot \sigma_u$.

*Proof.* We will use Lemma B.5, which connects sub-Gaussian and subexponential variables. Since the sub-Gaussian norms of $\|u_t\|_2$ for all $t$ are bounded by $\sigma_u$, we equivalently have

$$
\left\| \|u_t\|_2^2 \right\|_{\psi_1} \leq \sigma_u^2, \quad \forall t \geq 0.
$$

It follows that

$$
\left\| \|\boldsymbol{U}_t^{(\tau)}\|_2^2 \right\|_{\psi_1} = \left\| \sum_{i=0}^{\tau} \|u_{t-i}\|_2^2 \right\|_{\psi_1} \leq \sum_{i=0}^{\tau} \left\| \|u_{t-i}\|_2^2 \right\|_{\psi_1} \leq (\tau + 1) \sigma_u^2.
$$

We again hinge on Lemma B.5 to arrive at the conclusion. $\qquad\square$

**Lemma D.3.** *Suppose that Assumptions 2.4, 2.5, and 2.8 hold. Define* $\nu := \frac{\sqrt{M\tau} L_\phi \sigma_u}{\lambda}$. *Then, for a fixed* $Z \in \mathbb{R}^{r \times M}$ *such that* $\|Z\|_F = 1$, *we have*

$$
\mathbb{E}\left[ \mathbb{I}_{\pm}\{\boldsymbol{W}_t^{(\tau)} = 0\} \cdot \left\| Z\Phi(\boldsymbol{U}_t^{(\tau)}) \right\|_2 \right] = \Omega\left( \frac{(2(1-p)^\tau - 1) \cdot \lambda}{\nu^3} \right). \tag{35}
$$

*Proof.* We first analyze the sub-Gaussian norm of $\|Z\Phi(\boldsymbol{U}_t^{(\tau)})\|_2$. From (9) in Assumption 2.4 with $\Phi(0) = 0$, we have

$$
|\phi_i(\boldsymbol{U}_t^{(\tau)})| = |\phi_i(\boldsymbol{U}_t^{(\tau)}) - \phi_i(0)| \leq L_\phi \|\boldsymbol{U}_t^{(\tau)}\|_2. \tag{36}
$$

Due to Lemma D.2, it follows that

$$
\left\| |\phi_i(\boldsymbol{U}_t^{(\tau)})| \right\|_{\psi_2} \leq L_\phi \left\| \|\boldsymbol{U}_t^{(\tau)}\|_2 \right\|_{\psi_2} = L_\phi \sqrt{\tau + 1} \cdot \sigma_u
$$

Thus, one can obtain

$$
\left\| \|Z\Phi(\boldsymbol{U}_t^{(\tau)})\|_2 \right\|_{\psi_2} \leq \left\| \sum_{i=1}^{M} \|z_i \phi_i(\boldsymbol{U}_t^{(\tau)})\|_2 \right\|_{\psi_2} \leq \sum_{i=1}^{M} \left\| \|z_i \phi_i(\boldsymbol{U}_t^{(\tau)})\|_2 \right\|_{\psi_2}
$$

$$= \sum_{i=1}^{M} \|z_i\|_2 \left\| |\phi_i(\boldsymbol{U}_t^{(\tau)})| \right\|_{\psi_2} \leq \sqrt{M} L_\phi \sqrt{\tau+1} \cdot \sigma_u, \tag{37}$$

where $z_i$ is the $i^{\text{th}}$ column of $Z$. The inequalities follow from the triangle inequality and the Cauchy-Schwarz inequality.

Then, due to the property (24a), we have $\mathbb{E}\left[\|Z\Phi(\boldsymbol{U}_t^{(\tau)})\|_2^3\right] = O((\sqrt{M}\tau L_\phi \sigma_u)^3)$.

From (10) in Assumption 2.4, we have

$$\mathbb{E}\left[\|Z\Phi(\boldsymbol{U}_t^{(\tau)})\|_2^2\right] = \mathbb{E}[\text{trace}(Z^T Z \cdot \Phi(\boldsymbol{U}_t^{(\tau)})\Phi(\boldsymbol{U}_t^{(\tau)})^T)]$$

$$= \text{trace}(Z^T Z \cdot \mathbb{E}[\Phi(\boldsymbol{U}_t^{(\tau)})\Phi(\boldsymbol{U}_t^{(\tau)})^T]) \geq \lambda^2 \cdot \text{trace}(Z^T Z) = \lambda^2. \tag{38}$$

Note that

$$\mathbb{E}\left[\|Z\Phi(\boldsymbol{U}_t^{(\tau)})\|_2^2\right]^2 \leq \mathbb{E}\left[\|Z\Phi(\boldsymbol{U}_t^{(\tau)})\|_2\right] \cdot \mathbb{E}\left[\|Z\Phi(\boldsymbol{U}_t^{(\tau)})\|_2^3\right] \tag{39}$$

due to the Cauchy-Schwarz inequality. Combining the above two inequalities yields

$$\mathbb{E}\left[\|Z\Phi(\boldsymbol{U}_t^{(\tau)})\|_2\right] = \Omega\left(\frac{\lambda^4}{(\sqrt{M}\tau L_\phi \sigma_u)^3}\right). \tag{40}$$

Now, recall the relationship

$$\{\boldsymbol{W}_t^{(\tau)} = 0\} \supseteq \{w_{t-1} = 0, \ldots, w_{t-\tau} = 0\} \supseteq \{\xi_{t-1} = 0, \ldots, \xi_{t-\tau} = 0\},$$

which follows from Assumption 2.8. From the independence of $\xi_i$'s, we also have

$$\mathbb{P}(\xi_{t-1} = 0, \ldots, \xi_{t-\tau} = 0) = (1-p)^\tau > 0.5,$$

since $p < \frac{1}{2\tau}$. Then, one can write

$$\mathbb{E}\left[\mathbb{I}_\pm\{\boldsymbol{W}_t^{(\tau)} = 0\}\cdot\left\|Z\Phi(\boldsymbol{U}_t^{(\tau)})\right\|_2\right] \geq \mathbb{E}\left[\mathbb{I}_\pm\{\xi_{t-1} = 0, \ldots, \xi_{t-\tau} = 0\}\cdot\left\|Z\Phi(\boldsymbol{U}_t^{(\tau)})\right\|_2\right] \tag{41}$$

$$= \mathbb{E}\left[\mathbb{I}_\pm\{\xi_{t-1} = 0, \ldots, \xi_{t-\tau} = 0\}\right] \cdot \mathbb{E}\left[\left\|Z\Phi(\boldsymbol{U}_t^{(\tau)})\right\|_2\right]$$

$$\geq ((1-p)^\tau - (1 - (1-p)^\tau) \cdot \mathbb{E}\left[\left\|Z\Phi(\boldsymbol{U}_t^{(\tau)})\right\|_2\right]$$

$$= (2(1-p)^\tau - 1) \cdot \Omega\left(\frac{\lambda^4}{(\sqrt{M}\tau L_\phi \sigma_u)^3}\right). \tag{42}$$

We finally note that the term given in (42) is indeed positive since $(1-p)^\tau > 0.5$. Using the definition of $\nu$ completes the proof. □

We have defined $\nu$ in the above lemma. We will show that the value of $\nu$ is bounded below by a positive constant.

**Lemma D.4.** *Define $\nu := \frac{\sqrt{M}\tau L_\phi \sigma_u}{\lambda}$. Then, $\nu = \Omega(1)$.*

*Proof.* For any $Z \in \mathbb{R}^{r \times M}$ such that $\|Z\|_F = 1$, we have

$$\|Z\Phi(\boldsymbol{U}_t^{(\tau)})\|_2^2 \leq \|Z\|_F^2 \cdot \|\Phi(\boldsymbol{U}_t^{(\tau)})\|_2^2 = \|\Phi(\boldsymbol{U}_t^{(\tau)})\|_2^2 \leq M L_\phi^2 \|\boldsymbol{U}_t^{(\tau)}\|_2^2, \tag{43}$$

where the last inequality comes from (36). The expectation of the left-hand side of (43) is lower-bounded by $\lambda^2$ due to (39). Noting that the expectation and the $\psi_2$-norm of a nonnegative variable have the same order (see (24a)), the expectation of the right-hand side is upper-bounded by $O(M L_\phi^2 \tau \sigma_u^2)$ due to Lemma B.5. Thus, we have $\frac{M L_\phi^2 \tau \sigma_u^2}{\lambda} = \Omega(1)$; in other words, $\nu^2 = \Omega(1)$. This completes the proof. □

Now, we provide the crucial lemma to ensure that for a fixed $Z$, the term $\mathbb{I}_\pm\{\boldsymbol{W}_t^{(\tau)} = 0\}\cdot\left\|Z\Phi(\boldsymbol{U}_t^{(\tau)})\right\|_2$ is positive with probability at least $1 - \delta$.

**Lemma D.5.** *Suppose that Assumptions 2.4 and 2.5 hold. Define $\nu := \frac{\sqrt{M\tau}L_\phi\sigma_u}{\lambda}$. Given $\delta \in (0, 1]$, when*

$$T = \Omega\left(\frac{\tau\nu^8}{(2(1-p)^\tau - 1)^2}\log\left(\frac{1}{\delta}\right)\right), \tag{44}$$

*we have*

$$\sum_{t=\tau}^{T-1}\mathbb{I}_\pm\{\boldsymbol{W}_t^{(\tau)} = 0\}\cdot\left\|Z\Phi(\boldsymbol{U}_t^{(\tau)})\right\|_2 = \Omega\left(\frac{(2(1-p)^\tau - 1)\cdot\lambda T}{2\nu^3}\right) \tag{45}$$

*for a fixed $Z \in \mathbb{R}^{r\times M}$ such that $\|Z\|_F = 1$.*

*Proof.* Similar to (36) and (37), we have

$$\sum_{t=\tau}^{T-1}\|Z\Phi(\boldsymbol{U}_t^{(\tau)})\|_2 \le \sqrt{M}L_\phi\sum_{t=\tau}^{T-1}\sqrt{\sum_{i=0}^{\tau}\|u_{t-i}\|_2^2} \le \sqrt{M}L_\phi\sum_{t=\tau}^{T-1}\sum_{j=0}^{\tau}\|u_{t-j}\|_2. \tag{46}$$

Now, we define a Bernoulli variable $\boldsymbol{\Xi}_t^{(\tau)}$ such that $\{\boldsymbol{\Xi}_t^{(\tau)} = 0\} \Leftrightarrow \{\xi_{t-1} = 0, \ldots, \xi_{t-\tau} = 0\}$. From (41), we know that $\{\boldsymbol{W}_t^{(\tau)} = 0\} \supseteq \{\boldsymbol{\Xi}_t^{(\tau)} = 0\}$. Thus, it suffices to prove the desired result with $\boldsymbol{\Xi}_t^{(\tau)}$ in place of $\boldsymbol{W}_t^{(\tau)}$.

Consider the constants $A_1, \ldots, A_T > 0$. Then, for all $\theta \in \mathbb{R}$, we have

$$\underset{\substack{|a_t|\le A_t, \\ t=\tau,\ldots,T-1}}{\arg\max}\ \mathbb{E}\left[\exp\left(\theta\left(\sum_{t=\tau}^{T-1}a_t\cdot(\mathbb{I}_\pm\{\boldsymbol{\Xi}_t^{(\tau)} = 0\} - \mathbb{E}[\mathbb{I}_\pm\{\boldsymbol{\Xi}_t^{(\tau)} = 0\}])\right)^2\right)\right]$$

$$\subseteq \{\pm A_1\} \times \cdots \times \{\pm A_T\}, \tag{47}$$

since the function on the left-hand side is convex in $(a_1, \ldots, a_T)$ and the maximum of the convex function is attained at extreme points. Due to (46), substituting $\sum_{t=\tau}^{T-1}\|Z\Phi(\boldsymbol{U}_t^{(\tau)})\|_2$ into $a_t$ and $\sqrt{M}L_\phi(\tau + 1)\sum_{t=\tau}^{T-1}\sum_{j=0}^{\tau}\|u_{t-j}\|_2$ into $A_t$ in (47) yields

$$\left\|\sum_{t=\tau}^{T-1}\|Z\Phi(\boldsymbol{U}_t^{(\tau)})\|_2 \cdot \left(\mathbb{I}_\pm\{\boldsymbol{\Xi}_t^{(\tau)} = 0\} - \mathbb{E}[\mathbb{I}_\pm\{\boldsymbol{\Xi}_t^{(\tau)} = 0\}]\right)\right\|_{\psi_2}$$

$$\le \left\|\sqrt{M}L_\phi(\tau + 1)\sum_{t=\tau}^{T-1}\sum_{j=0}^{\tau}\|u_{t-j}\|_2 \cdot \left(\mathbb{I}_\pm\{\boldsymbol{\Xi}_t^{(\tau)} = 0\} - \mathbb{E}[\mathbb{I}_\pm\{\boldsymbol{\Xi}_t^{(\tau)} = 0\}]\right)\right\|_{\psi_2} \tag{48}$$

considering that $\boldsymbol{\Xi}_t^{(\tau)}$ is independent of any other variables and the expectation of $\mathbb{I}_\pm\{\boldsymbol{\Xi}_t^{(\tau)} = 0\} - \mathbb{E}[\mathbb{I}_\pm\{\boldsymbol{\Xi}_t^{(\tau)} = 0\}]$ is zero, in which case the sub-Gaussian norm can be determined by (24c).

Now, we analyze the right-hand side of (48). For simplicity, we define

$$\Xi_t := \begin{cases} 0, & t = 0, \ldots, \tau - 1, \\ \mathbb{I}_\pm\{\boldsymbol{\Xi}_t^{(\tau)} = 0\} - \mathbb{E}\big[\mathbb{I}_\pm\{\boldsymbol{\Xi}_t^{(\tau)} = 0\}\big], & t = \tau, \ldots, T - 1, \\ 0, & t = T, \ldots, T + \tau - 1. \end{cases}$$

Then, we have

$$\sum_{t=\tau}^{T-1}\sum_{j=0}^{\tau}\|u_{t-j}\|_2 \cdot \Xi_t = \sum_{t=0}^{T-1}\left(\sum_{j=t}^{t+\tau}\Xi_j\right)\cdot\|u_t\|_2. \tag{49}$$

For all $t$, we have

$$\left\|\sum_{t=0}^{T-1}\left(\sum_{j=t}^{t+\tau}\Xi_j\right)\cdot\|u_t\|_2\right\|_{\psi_2} \le (\tau + 1)\sigma_u$$

due to Assumption 2.5. Given the filtration $\mathcal{F}^i = \boldsymbol{\sigma}\{\Xi_t : t = 0, \ldots, T-1\}$ and considering that $\mathbb{E}[\Xi_t] = 0$, we can apply the property (24c) to obtain

$$\mathbb{E}\left[\exp\left(\theta\sum_{t=0}^{T-1}\left(\sum_{j=t}^{t+\tau}\Xi_j\right)\cdot\|u_t\|_2\right)\right] \leq \mathbb{E}\left[\mathbb{E}\left[\exp\left(\theta\sum_{t=0}^{T-1}\left(\sum_{j=t}^{t+\tau}\Xi_j\right)\cdot\|u_t\|_2\right)\right]\,\Big|\,\mathcal{F}^i\right]$$

$$\leq \prod_{t=0}^{T-1}\exp(\theta^2(\tau+1)^2\sigma_u^2) = \exp(\theta^2 T(\tau+1)^2\sigma_u^2), \quad (50)$$

for all $\theta \in \mathbb{R}$, which implies that the mean-zero variable (49) is sub-Gaussian and its norm is $O(\sqrt{T}(\tau+1)\sigma_u)$. In turn, due to (48), we arrive at

$$\left\|\sum_{t=\tau}^{T-1} Z\Phi(\boldsymbol{U}_t^{(\tau)})\cdot\Xi_t\right\|_{\psi_2} = O(\sqrt{TM}L_\phi(\tau+1)\sigma_u). \quad (51)$$

Finally, we can apply the property (25b) to obtain

$$\mathbb{P}\left(\sum_{t=\tau}^{T-1} Z\Phi(\boldsymbol{U}_t^{(\tau)})\cdot\Xi_t > -\Omega\left(\frac{(2(1-p)^\tau - 1)\cdot\lambda T}{2\nu^3}\right)\right)$$

$$\geq 1 - \exp\left(-\Omega\left(\frac{(2(1-p)^\tau - 1)^2\lambda^2 T^2}{(\sqrt{TM}L_\phi(\tau+1)\sigma_u)^2\nu^6}\right)\right)$$

$$= 1 - \exp\left(-\Omega\left(\frac{(2(1-p)^\tau - 1)^2\cdot T}{\tau\nu^8}\right)\right).$$

We derive from (42) that

$$\mathbb{E}\left[\sum_{t=\tau}^{T-1} Z\Phi(\boldsymbol{U}_t^{(\tau)})\cdot\mathbb{I}_\pm\{\boldsymbol{\Xi}_t^{(\tau)} = 0\}\right] = \Omega\left(\frac{(2(1-p)^\tau - 1)\cdot\lambda T}{\nu^3}\right).$$

Since $\Xi_t = \mathbb{I}_\pm\{\boldsymbol{\Xi}_t^{(\tau)} = 0\} - \mathbb{E}\left[\mathbb{I}_\pm\{\boldsymbol{\Xi}_t^{(\tau)} = 0\}\right]$, we arrive at

$$\mathbb{P}\left(\sum_{t=\tau}^{T-1} Z\Phi(\boldsymbol{U}_t^{(\tau)})\cdot\mathbb{I}_\pm\{\boldsymbol{\Xi}_t^{(\tau)} = 0\} > \Omega\left(\frac{(2(1-p)^\tau - 1)\cdot\lambda T}{2\nu^3}\right)\right)$$

$$\geq 1 - \exp\left(-\Omega\left(\frac{(2(1-p)^\tau - 1)^2\cdot T}{\tau\nu^8}\right)\right). \quad (52)$$

Since we have $\{\boldsymbol{W}_t^{(\tau)} = 0\} \supseteq \{\boldsymbol{\Xi}_t^{(\tau)} = 0\}$, establishing a lower bound of $1 - \delta$ on the right-hand side of (52) suffices to conclude the proof. $\qquad\square$

We now study the effect of perturbing $Z \in \mathbb{R}^{r\times M}$.

**Lemma D.6.** *Suppose that Assumptions 2.4 and 2.5 hold. Given $\delta \in (0,1]$, when $T = \Omega(\log(2/\delta))$, the inequality*

$$\sum_{t=\tau}^{T-1}\mathbb{I}_\pm\{\boldsymbol{W}_t^{(\tau)} = 0\}\cdot\left\|Z\Phi(\boldsymbol{U}_t^{(\tau)})\right\|_2 - \sum_{t=\tau}^{T-1}\mathbb{I}_\pm\{\boldsymbol{W}_t^{(\tau)} = 0\}\cdot\left\|\tilde{Z}\Phi(\boldsymbol{U}_t^{(\tau)})\right\|_2 \geq -O(T\|Z-\tilde{Z}\|_F L_\phi\sqrt{M}\tau\sigma_u)$$

*holds for every $Z, \tilde{Z} \in \mathbb{R}^{r\times M}$ with probability at least $1 - \frac{\delta}{2}$.*

*Proof.* For simplicity, we define $\bar{f}_t(Z) := \mathbb{I}_\pm\{\boldsymbol{W}_t^{(\tau)} = 0\}\cdot\left\|Z\Phi(\boldsymbol{U}_t^{(\tau)})\right\|_2$. For $Z, \tilde{Z} \in \mathbb{R}^{r\times M}$, we have

$$\sum_{t=\tau}^{T-1}\bar{f}_t(Z) - \sum_{t=\tau}^{T-1}\bar{f}_t(\tilde{Z}) \geq -\sum_{t=\tau}^{T-1}\|(Z-\tilde{Z})\Phi(\boldsymbol{U}_t^{(\tau)})\|_2$$

$$\geq -\sum_{t=\tau}^{T-1} \|Z - \tilde{Z}\|_F \cdot L_\phi \sqrt{M} \cdot \sum_{j=0}^{\tau} \|u_{t-j}\|_2$$

$$\geq -\sum_{t=0}^{T-1} \|Z - \tilde{Z}\|_F L_\phi \sqrt{M}(\tau + 1)\|u_t\|_2, \qquad (53)$$

where the first inequality is due to the triangle inequality and the second comes from (46). Assumption 2.5 tells that $\|\|u_t\|_2\| \leq \sigma_u$ and thus, we have

$$\left\| \|Z - \tilde{Z}\|_F L_\phi \sqrt{M}(\tau + 1)(\|u_t\|_2 - \mathbb{E}[\|u_t\|_2]) \right\|_{\psi_2} = \|Z - \tilde{Z}\|_F L_\phi \sqrt{M}(\tau + 1) \cdot O(\sigma_u)$$

due to Lemma B.2. In turn, due to Lemma B.3 and the independence of control inputs, we have

$$\left\| \sum_{t=0}^{T-1} \|Z - \tilde{Z}\|_F L_\phi \sqrt{M}(\tau + 1)(\|u_t\|_2 - \mathbb{E}[\|u_t\|_2]) \right\|_{\psi_2} = \|Z - \tilde{Z}\|_F L_\phi \sqrt{M}(\tau + 1) \cdot O(\sqrt{T}\sigma_u).$$

Using the property (25a), one can obtain

$$\mathbb{P}\left( \sum_{t=0}^{T-1} \|Z - \tilde{Z}\|_F L_\phi \sqrt{M}(\tau + 1)(\|u_t\|_2 - \mathbb{E}[\|u_t\|_2]) \leq \|Z - \tilde{Z}\|_F L_\phi \sqrt{M}(\tau + 1) \cdot O(T\sigma_u) \right)$$

$$\geq 1 - \exp\left( -\Omega\left( \frac{T^2\|Z - \tilde{Z}\|_F^2 L_\phi^2 M(\tau + 1)^2 \sigma_u^2}{T\|Z - \tilde{Z}\|_F^2 L_\phi^2 M(\tau + 1)^2 \sigma_u^2} \right) \right) = 1 - \exp(-\Omega(T)).$$

Note that $\mathbb{E}[\|u_t\|_2] = O(\sigma_u)$ due to (24a). Thus, we can write

$$\mathbb{P}\left( \sum_{t=0}^{T-1} \|Z - \tilde{Z}\|_F L_\phi \sqrt{M}(\tau + 1)\|u_t\|_2 \leq 2\|Z - \tilde{Z}\|_F L_\phi \sqrt{M}(\tau + 1) \cdot O(T\sigma_u) \right) \geq 1 - \exp(-\Omega(T)).$$
$$(54)$$

When $T = \Omega(\log(2/\delta))$, the probability in (54) is lower-bounded by $1 - \frac{\delta}{2}$. Considering the lower bound of (53) completes the proof. $\qquad\square$

Now, we will achieve that the inequality (31) in Theorem D.1 holds for all $Z \in \mathbb{R}^{r \times M}$ such that $\|Z\|_F = 1$, after finite time. To take advantage of Lemma D.6, which states the difference of $\sum_t \bar{f}_t(Z)$ depending on $Z$, we introduce the important lemma presented in Vershynin (2010).

**Lemma D.7** (Covering number of the sphere). *Define* $\mathbb{S}^{r \times M-1} := \{Z \in \mathbb{R}^{r \times M} : \|Z\|_F = 1\}$. *For* $\epsilon > 0$, *consider a subset* $\mathcal{N}_\epsilon$ *of* $\mathbb{S}^{r \times M-1}$, *such that for all* $Z \in \mathbb{S}^{r \times M-1}$, *there exists some point* $\tilde{Z} \in \mathcal{N}_\epsilon$ *satisfying* $\|Z - \tilde{Z}\|_2 \leq \epsilon$. *The minimal cardinality of such a subset is called the covering number of the sphere and is upper-bounded by* $(1 + \frac{2}{\epsilon})^{rM}$.

The covering number argument states that if you select $(1 + \frac{2}{\epsilon})^{rM}$ number of points which achieve the sufficient positiveness of $\sum_t \bar{f}_t(Z)$, and show that the difference of $\sum_t \bar{f}_t(Z)$ is small enough within the distance $\epsilon$, then all the points will achieve the positiveness of $\sum_t \bar{f}_t(Z)$.

**Theorem D.8.** *Suppose that Assumptions 2.4 and 2.5 hold. Consider* $\nu := \frac{\sqrt{M}\tau L_\phi \sigma_u}{\lambda}$ *and* $\mathbb{S}^{r \times M-1} := \{Z \in \mathbb{R}^{r \times M} : \|Z\|_F = 1\}$. *Given* $\delta \in (0, 1]$, *when*

$$T = \Omega\left( \frac{\tau\nu^8}{(2(1-p)^\tau - 1)^2} \left[ rM \log\left( \frac{\tau\nu}{2(1-p)^\tau - 1} \right) + \log\left( \frac{1}{\delta} \right) \right] \right), \qquad (55)$$

*we have*

$$\sum_{t=\tau}^{T-1} \mathbb{I}_\pm\{\boldsymbol{W}_t^{(\tau)} = 0\} \cdot \left\| Z\Phi(\boldsymbol{U}_t^{(\tau)}) \right\|_2 = \Omega\left( \frac{(2(1-p)^\tau - 1) \cdot \lambda T}{4\nu^3} \right) > 0, \quad \forall Z \in \mathbb{S}^{r \times M-1} \qquad (56)$$

*with probability at least* $1 - \delta$.

*Proof.* As in the previous lemma, we define $\bar{f}_t(Z) := \mathbb{I}_{\pm}\{\boldsymbol{W}_t^{(\tau)} = 0\} \cdot \left\|Z\Phi(\boldsymbol{U}_t^{(\tau)})\right\|_2$. Also, define $\epsilon^* = \frac{1}{4}O\left(\frac{2(1-p)^{\tau}-1}{\tau^{1/2}\nu^4}\right)$. From Lemma D.6, for all $Z, \tilde{Z} \in \mathbb{S}^{r \times M-1}$ satisfying $\|Z - \tilde{Z}\|_F \leq \epsilon^*$, we have

$$\sum_{t=\tau}^{T-1} \bar{f}_t(Z) - \sum_{t=\tau}^{T-1} \bar{f}_t(\tilde{Z}) \geq -O(T\epsilon^* L_\phi \sqrt{M}\tau\sigma_u) \geq -\frac{1}{4}O\left(T \cdot \frac{2(1-p)^{\tau}-1}{\tau^{1/2}\nu^3}\frac{L_\phi\sqrt{M}\tau\sigma_u}{\nu}\right)$$

$$= -\frac{1}{4}O\left(\frac{(2(1-p)^{\tau}-1)\lambda T}{\nu^3}\right). \tag{57}$$

with probability at least $1 - \frac{\delta}{2}$, when $T = \Omega(\log(2/\delta))$. If we select $(1 + \frac{2}{\epsilon^*})^{rM}$ points $\{Z_1, \ldots, Z_{(1+\frac{2}{\epsilon^*})^{rM}}\}$ satisfying (45) with probability at least $1 - \frac{\delta}{2 \cdot (1+\frac{2}{\epsilon^*})^{rM}}$, then it follows that

$$\sum_{t=\tau}^{T-1} \bar{f}_t(Z) = \frac{1}{2}\Omega\left(\frac{(2(1-p)^{\tau}-1)\lambda T}{\nu^3}\right), \quad \forall Z \in \hat{Z} = \{Z_1, \ldots, Z_{(1+\frac{2}{\epsilon^*})^{rM}}\} \tag{58}$$

with probability at least $1 - \frac{\delta}{2}$. Then, due to Lemma D.7, every point in $\mathbb{S}^{r \times M-1}$ is within a distance of $\epsilon^*$ from at least one point in $\hat{Z}$. In turn, by (57), we have

$$\sum_{t=\tau}^{T-1} \bar{f}_t(Z) \geq \frac{1}{4}\Omega\left(\frac{(2(1-p)^{\tau}-1)\lambda T}{\nu^3}\right) > 0, \quad \forall Z \in \mathbb{S}^{r \times M-1} \tag{59}$$

holds with probability at least $1 - \delta$. Thus, we replace $\delta$ in (44) with $\frac{\delta}{2 \cdot (1+\frac{2}{\epsilon^*})^{rM}}$ to arrive at

$$T = \Omega\left(\frac{\tau\nu^8}{(2(1-p)^{\tau}-1)^2}\log\left(\frac{2(1+\frac{2}{\epsilon^*})^{rM}}{\delta}\right)\right)$$

$$= \Omega\left(\frac{\tau\nu^8}{(2(1-p)^{\tau}-1)^2}\left[rM\log\left(1 + \frac{2}{\epsilon^*}\right) + \log\left(\frac{1}{\delta}\right)\right]\right)$$

$$= \Omega\left(\frac{\tau\nu^8}{(2(1-p)^{\tau}-1)^2}\left[rM\log\left(\frac{\tau\nu}{2(1-p)^{\tau}-1}\right) + \log\left(\frac{1}{\delta}\right)\right]\right), \tag{60}$$

where we leveraged Lemma D.4 for the last equality. Note that $T = \Omega(\log(2/\delta))$ required for (57) is automatically satisfied with the recovery time (60). This completes the proof. $\square$

In Theorem D.8, we achieve that $\sum_t \bar{f}_t(Z)$ is sufficiently positive after the recovery time given in (55). Thus, we arrive at the conclusion that when $\boldsymbol{x}_t^{(\tau)} = 0$ and $\boldsymbol{\epsilon}_t = 0$ for all $t$, $G^*$ is the unique solution to the $\ell_2$-norm estimator (12) after finite time due to Lemma D.1.

We will now generalize for the case of nonzero $\boldsymbol{x}_t^{(\tau)}$ and $\boldsymbol{\epsilon}_t$. Before presenting the main theorem, we provide the following useful lemma.

**Lemma D.9.** *Suppose that Assumptions 2.1, 2.5, and 2.6 hold. Given $\delta \in (0, 1]$, when $T = \Omega(\log(1/\delta))$,*

$$\sum_{t=0}^{T-\tau-1} \|x_t\|_2 = O\left(\frac{(\sigma_u + \sigma_w)}{1 - \rho} \cdot T\right) \tag{61}$$

*holds with probability at least $1 - \delta$.*

*Proof.* Due to the inequality (4) in Assumption 2.1, we have

$$\|x_t\|_2 = \|f(x_{t-1}, u_{t-1}, w_{t-1})\|_2 = \cdots = \|f(f(\cdots f(f(x_0, u_0, w_0), u_1, w_1), \cdots), \cdots), u_{t-1}, w_{t-1})\|_2$$

$$= \|f(f(f(x_0, u_0, w_0), u_1, w_1), \cdots), \cdots), u_{t-2}, w_{t-2}, u_{t-1}, w_{t-1})$$

$$- f(f(f(0, 0, 0), 0, 0), \cdots), \cdots), 0, 0), 0, 0)\|_2$$

$$\leq \|f(f(f(x_0, u_0, w_0), u_1, w_1), \cdots), \cdots), u_{t-2}, w_{t-2}, u_{t-1}, w_{t-1})$$

$$- f(f(f(x_0, u_0, w_0), u_1, w_1), \cdots), \cdots), u_{t-2}, w_{t-2}, 0, 0)\|_2$$

$$+ \|f(f(f(x_0, u_0, w_0), u_1, w_1), \cdots), \cdots), u_{t-2}, w_{t-2}), 0, 0)$$
$$- f(f(f(x_0, u_0, w_0), u_1, w_1), \cdots), \cdots), 0, 0), 0, 0)\|_2$$
$$+ \cdots$$
$$+ \|f(f(f(x_0, u_0, w_0), 0, 0), \cdots), \cdots), 0, 0), 0, 0)$$
$$- f(f(f(0, 0, 0), 0, 0), \cdots), \cdots), 0, 0), 0, 0)\|_2 \tag{62}$$

where the equality in the second line comes from $f(0, 0, 0) = 0$ and the inequality is due to the triangle inequality. By Assumption 2.1, the terms in (62) are bounded by

$$C\rho(\|u_{t-1}\|_2 + \|w_{t-1}\|_2), \quad C\rho^2(\|u_{t-2}\|_2 + \|w_{t-2}\|_2), \ldots,$$
$$C\rho^{t-1}(\|u_1\|_2 + \|w_1\|_2), \quad C\rho^t(\|x_0\|_2 + \|u_0\|_2 + \|w_0\|_2).$$

Thus, we have

$$\|x_t\|_2 \le C\rho^t \|x_0\|_2 + C\sum_{i=0}^{t-1} \rho^{t-i}(\|u_i\|_2 + \|w_i\|_2).$$

Summing up for $t = 0, \ldots, T - \tau - 1$ yields

$$\sum_{t=0}^{T-\tau-1} \|x_t\|_2 \le C \sum_{t=0}^{T-\tau-1} \rho^t \|x_0\|_2 + C \sum_{t=0}^{T-\tau-1} \sum_{i=0}^{t-1} \rho^{t-i}(\|u_i\|_2 + \|w_i\|_2)$$
$$< C \sum_{t=0}^{\infty} \rho^t \|x_0\|_2 + C \sum_{t=0}^{\infty} \rho^t \sum_{i=0}^{T-\tau-2} (\|u_i\|_2 + \|w_i\|_2)$$
$$= \frac{C}{1-\rho} \Big[ \|x_0\|_2 + \sum_{i=0}^{T-\tau-2} \|w_i\|_2 + \sum_{i=0}^{T-\tau-2} \|u_i\|_2 \Big] \tag{63}$$

Consider that

$$\mathbb{E}\Big[ \exp\Big( \theta \Big[ \|x_0\|_2 - \mathbb{E}[\|x_0\|_2] + \sum_{i=0}^{T-\tau-2} \|w_i\|_2 - \mathbb{E}[\|w_i\|_2] \Big] \Big) \Big]$$

$$= \mathbb{E}\Big[ \mathbb{E}\Big[ \exp\Big( \theta \Big[ \|x_0\|_2 - \mathbb{E}[\|x_0\|_2] + \sum_{i=0}^{T-\tau-2} \|w_i\|_2 - \mathbb{E}[\|w_i\|_2] \Big] \Big) \Big| \mathcal{F}_{T-\tau-2} \Big] \Big]$$

$$= \mathbb{E}\Big[ \mathbb{E}\Big[ \exp\big( \theta(\|w_{T-2}\|_2 - \mathbb{E}[\|w_{T-2}\|_2]) \big) \big| \mathcal{F}_{T-\tau-2} \Big]$$
$$\times \exp\Big( \theta \Big[ \|x_0\|_2 - \mathbb{E}[\|x_0\|_2] + \sum_{i=0}^{T-\tau-3} \|w_i\|_2 - \mathbb{E}[\|w_i\|_2] \Big] \Big) \Big]$$

$$\le \exp(\theta^2 \cdot O(\sigma_w^2)) \cdot \mathbb{E}\Big[ \exp\Big( \theta \Big[ \|x_0\|_2 - \mathbb{E}[\|x_0\|_2] + \sum_{i=0}^{T-\tau-3} \|w_i\|_2 - \mathbb{E}[\|w_i\|_2] \Big] \Big) \Big]$$

$$\le \cdots \le \exp(\theta^2 \cdot O(T\sigma_w^2))$$

for all $\theta \in \mathbb{R}$, where the inequalities come from applying Lemma B.2 to Assumption 2.6. Thus, the sub-Gaussian norm of $\|x_0\|_2 - \mathbb{E}[\|x_0\|_2] + \sum_{i=0}^{T-\tau-2} \|w_i\|_2 - \mathbb{E}[\|w_i\|_2]$ is $O(\sqrt{T}\sigma_w)$. Furthermore, since the sub-Gaussian norm of $\|u_i\|_2$ is $\sigma_u$ due to Assumption 2.5, the sub-Gaussian norm of $\sum_{i=0}^{T-\tau-2} \|u_i\|_2 - \mathbb{E}[\sum_{i=0}^{T-\tau-2} \|u_i\|_2]$ is $O(\sqrt{T}\sigma_u)$ by applying Lemmas B.2 and B.3.

Denote the term in (63) as $S_T$. Considering the aforementioned sub-Gaussian norms, the sub-Gaussian norm of $S_T - \mathbb{E}[S_T]$ is $O(\sqrt{T}\frac{\sigma_w + \sigma_u}{1-\rho})$ due to the triangle inequality and the homogeneity. Due to the property (25a), one arrives at

$$\mathbb{P}\left( S_T - \mathbb{E}[S_T] \le O\left( \frac{\sigma_w + \sigma_u}{1-\rho} \cdot T \right) \right) \ge 1 - \exp\left( -\Omega\left( \frac{T^2(\sigma_w + \sigma_u)^2/(1-\rho)^2}{T(\sigma_w + \sigma_u)^2/(1-\rho)^2} \right) \right)$$
$$= 1 - \exp(-\Omega(T)) \tag{64}$$

We additionally have

$$\mathbb{E}\left[S_T\right] = \frac{C}{1-\rho}\left[\mathbb{E}[\|x_0\|_2] + \sum_{i=0}^{T-\tau-2}\left(\mathbb{E}[\|u_i\|_2] + \mathbb{E}[\|w_i\|_2]\right)\right]$$

$$\leq \frac{C}{1-\rho}[O(T\sigma_w + T\sigma_u)] = O\left(\frac{\sigma_w + \sigma_u}{1-\rho} \cdot T\right), \tag{65}$$

where the last inequality is obtained by applying the property (24a) to Assumptions 2.5 and 2.6. Combining (64) and (65) yields

$$\mathbb{P}\left(S_T \leq 2 \cdot O\left(\frac{\sigma_w + \sigma_u}{1-\rho} \cdot T\right)\right) \geq 1 - \exp(-\Omega(T)) \geq 1 - \delta \tag{66}$$

when $T = \Omega(\log(1/\delta))$. Recall from (63) that $\sum_{t=0}^{T-\tau-1}\|x_t\|_2$ is bounded above by $S_T$. This completes the proof. □

Now, we present our main theorem, which states that the estimation error to identify $G^*$ in the system (8) is bounded by $O(\rho^\tau)$ when using the $\ell_2$-norm estimator.

**Theorem D.10** (Restatement of Theorem 3.1). *Suppose that Assumptions 2.1, 2.4, 2.5, 2.6, and 2.8 hold, and that the approximation error vector satisfies $\|\epsilon_t\|_2 \leq \bar{\epsilon}$ for all t. Consider $\nu := \frac{\sqrt{M\tau}L_\phi\sigma_u}{\lambda}$. Let $G^*$ be the true matrix governing the system (8) and $\hat{G}_T$ denote a solution to the $\ell_2$-norm estimator given in (12). Given $\delta \in (0, 1]$, when*

$$T = \Omega\left(\frac{\tau\nu^8}{(2(1-p)^\tau - 1)^2}\left[rM\log\left(\frac{\tau\nu}{2(1-p)^\tau - 1}\right) + \log\left(\frac{1}{\delta}\right)\right]\right), \tag{67}$$

*we have*

$$\|G^* - \hat{G}_T\|_F = O\left(\left(\frac{\rho^\tau L}{\lambda} \cdot \frac{\sigma_u + \sigma_w}{1-\rho} + \frac{\bar{\epsilon}}{\lambda}\right) \cdot \frac{\nu^3}{2(1-p)^\tau - 1}\right) \tag{68}$$

*with probability at least $1 - \delta$.*

*Proof.* The optimality of $\hat{G}_T$ to the $\ell_2$-norm estimator (12) for the system (8) yields

$$\hat{G}_T = \arg\min_G \sum_{t=\tau}^{T-1}\left\|(G^* - G) \cdot \Phi(U_t^{(\tau)}) + W_t^{(\tau)} + x_t^{(\tau)} + \epsilon_t\right\|_2,$$

which implies that

$$\sum_{t=\tau}^{T-1}\|(G^* - \hat{G}_T)\Phi(U_t^{(\tau)}) + W_t^{(\tau)}\|_2 - \|x_t^{(\tau)} + \epsilon_t\|_2 \tag{69}$$

$$\leq \sum_{t=\tau}^{T-1}\|(G^* - \hat{G}_T)\Phi(U_t^{(\tau)}) + W_t^{(\tau)} + x_t^{(\tau)} + \epsilon_t\|_2 \leq \sum_{t=\tau}^{T-1}\|W_t^{(\tau)} + x_t^{(\tau)} + \epsilon_t\|_2 \tag{70}$$

$$\leq \sum_{t=\tau}^{T-1}\|W_t^{(\tau)}\|_2 + \|x_t^{(\tau)} + \epsilon_t\|_2, \tag{71}$$

where (70) uses the optimality of $\hat{G}_T$ and the other inequalities are from the triangle inequality. By rearranging, we have

$$\sum_{t=\tau}^{T-1}\|(G^* - \hat{G}_T)\Phi(U_t^{(\tau)}) + W_t^{(\tau)}\|_2 - \|W_t^{(\tau)}\|_2 \leq 2\sum_{t=\tau}^{T-1}\|x_t^{(\tau)}\|_2 + \|\epsilon_t\|_2, \tag{72}$$

where the inequality is by (69) and (71). Recall from Lemma 2.3 that $\|x_t^{(\tau)}\|_2 \leq CL\rho^\tau\|x_{t-\tau}\|_2$. Then, we can establish that

$$2\sum_{t=\tau}^{T-1}\|x_t^{(\tau)}\|_2 \leq 2\sum_{t=\tau}^{T-1}CL\rho^\tau\|x_{t-\tau}\|_2 = 2\sum_{t=0}^{T-\tau-1}CL\rho^\tau\|x_t\|_2. \tag{73}$$

Given the time (67), the right-hand side of (72) is upper bounded by

$$2 \cdot O\left(\left(\frac{CL\rho^\tau(\sigma_u + \sigma_w)}{1-\rho} + \bar{\epsilon}\right)T\right)$$

with probability at least $1 - \frac{\delta}{2}$, which follows from Lemma D.9 and $\|\epsilon_t\|_2 \leq \bar{\epsilon}$.

We now aim to lower bound the left-hand side of (72) given the time (67). Inspired by (33), we have

$$\sum_{t=\tau}^{T-1} \|(G^* - \hat{G}_T)\Phi(U_t^{(\tau)}) + W_t^{(\tau)}\|_2 - \|W_t^{(\tau)}\|_2 \geq \sum_{t=\tau}^{T-1} \mathbb{I}_\pm\{W_t^{(\tau)} = 0\}\cdot\|(G^* - \hat{G}_T)\cdot\Phi(U_t^{(\tau)})\|_2$$

$$= \|G^* - \hat{G}_T\|_F \cdot \mathbb{I}_\pm\{W_t^{(\tau)} = 0\}\cdot\left\|\frac{G^* - \hat{G}_T}{\|G^* - \hat{G}_T\|_F} \cdot \Phi(U_t^{(\tau)})\right\|_2$$

$$= \|G^* - \hat{G}_T\|_F \cdot \Omega\left(\frac{(2(1-p)^\tau - 1)\cdot\lambda T}{4\nu^3}\right)$$

where the first equality comes from the homogeneity of the $\ell_2$-norm, and the second equality holds for any $G^* - \hat{G}_T$ with probability at least $1 - \frac{\delta}{2}$ due to Theorem D.8.

Thus, with probability at least $1 - \delta$, we have

$$\|G^* - \hat{G}_T\|_F \cdot \Omega\left(\frac{(2(1-p)^\tau - 1)\cdot\lambda T}{4\nu^3}\right) \leq 2 \cdot O\left(\left(\frac{CL\rho^\tau(\sigma_u + \sigma_w)}{1-\rho} + \bar{\epsilon}\right)T\right),$$

which can be rearranged to

$$\|G^* - \hat{G}_T\|_F = O\left(\left(\frac{\rho^\tau L(\sigma_u + \sigma_w)}{1-\rho} + \bar{\epsilon}\right) \cdot \frac{\nu^3}{(2(1-p)^\tau - 1)\lambda}\right).$$

This completes the proof. $\qquad\square$

# E  PROOF OF THEOREM 3.5

*Proof.* Let $M_T$ denote the maximum consecutive attack-free time length during $t = 0, \ldots, T-1$ under the attack probability $\frac{1}{2\tau+1}$, which satisfies Assumption 2.8. Then, due to the union bound, we have

$$\mathbb{P}(M_T \geq l) \leq \sum_{t=0}^{T-1} \mathbb{P}(\text{no attack occurs from time } t \text{ to } t+l) = \sum_{t=0}^{T-1}\left(1 - \frac{1}{2\tau+1}\right)^l. \qquad (74)$$

For the right-hand side to be less than $\delta$, we have

$$T\left(1 - \frac{1}{2\tau+1}\right)^l < \delta \iff l \geq \frac{\log(T/\delta)}{-\log\left(1 - \frac{1}{2\tau+1}\right)}.$$

Since we have $-\log(1-x) = x + \frac{x^2}{2} + \frac{x^3}{3} + \cdots \leq x + x^2 + x^3 + \cdots = \frac{x}{1-x} < 2x$ for $|x| < \frac{1}{2}$, it follows that

$$l \geq \frac{\log(T/\delta)}{\frac{2}{2\tau+1}} \geq \tau\log\left(\frac{T}{\delta}\right).$$

Thus, we arrive at

$$\mathbb{P}\left(M_T < \tau\log\left(\frac{T}{\delta}\right)\right) \geq 1 - \delta \qquad (75)$$

Now, consider the following functions $f, g : \mathbb{R} \to \mathbb{R}$:

$$f(x, u, w) = \rho(x + u + w), \quad g(x, u) = L(x + u), \qquad (76)$$

which satisfies Assumption 2.1. Then, as in (5), the observation $y_t$ can be written as

$$y_t = g(f(\cdots f(f(x_{t-\tau}, u_{t-\tau}, w_{t-\tau}), u_{t-\tau+1}, w_{t-\tau+1}), \cdots, u_{t-1}, w_{t-1}), u_t)$$
$$= L(\rho(\cdots \rho(\rho(x_{t-\tau} + u_{t-\tau} + w_{t-\tau}) + u_{t-\tau+1} + w_{t-\tau+1})\cdots + u_{t-1} + w_{t-1}) + u_t) \quad (77)$$

Suppose that control inputs $u_t$ are chosen independently from $\{-1, 1\}$ with equal probability for all $t = 0, \ldots, T - 1$, which satisfies Assumption 2.5. Given a finite $\sigma_w = \left(\frac{1}{\rho}\right)^{\Omega(\tau \log(T/\delta))}$, start the system with $x_0 = \sigma_w$ and let the disturbance $w_t$ also be $\sigma_w$ whenever the attack occurs at each time $t$, which satisfies Assumption 2.6. Note that the dynamics $f$ shrinks the system by a factor of $\rho$. Then, considering (75), one can ensure that adversarial attacks yield $x_t \geq 1$ for all $t = 0, \ldots, T - 1$ with probability at least $1 - \delta$. In this case, we can also rewrite (77) as:

$$y_t = L(\rho(\cdots \rho(\beta(\rho(x_{t-\tau} + u_{t-\tau} + w_{t-\tau})) + u_{t-\tau+1} + w_{t-\tau+1})\cdots + u_{t-1} + w_{t-1}) + u_t), \tag{78}$$

where

$$\beta(x) = \begin{cases} \dfrac{\tanh(x)}{\tanh(1)}, & \text{if } -1 \leq x \leq 1, \\ x, & \text{otherwise,} \end{cases} \tag{79}$$

which is a Lipschitz continuous function. The expressions in (77) and (78) have exactly the same function value since $\rho(x_{t-\tau} + u_{t-\tau} + w_{t-\tau}) = x_{t-\tau+1} \geq 1$ under adversarial attacks. In other words, one cannot distinguish between the two expressions (77) and (78). For each expression, the natural input-output mapping as in (6) would be

$$L(\rho(\cdots \rho(\rho(u_{t-\tau}) + u_{t-\tau+1})\cdots + u_{t-1}) + u_t) \quad \text{and} \tag{80a}$$
$$L(\rho(\cdots \rho(\beta(\rho(u_{t-\tau})) + u_{t-\tau+1})\cdots + u_{t-1}) + u_t), \tag{80b}$$

respectively. Define the constant

$$c := \left| 1 - \frac{\tanh(\rho)}{\rho \tanh(1)} \right|,$$

where one has $0 < c < 1$ under $0 < \rho < 1$. Then, the absolute difference of (80a) and (80b) is calculated as

$$L\rho^{\tau-1}|\rho u_{t-\tau} - \beta(\rho u_{t-\tau})| = L\rho^{\tau-1}|\rho - \beta(\rho)| = L\rho^\tau c,$$

since $u_{t-\tau}$ is selected from $-1$ and $1$, and $\beta(x)$ is an odd function. Now, let the basis function be

$$\Phi(\boldsymbol{U}_t^{(\tau)}) = \begin{bmatrix} L(\rho(\cdots \rho(\rho(u_{t-\tau}) + u_{t-\tau+1})\cdots + u_{t-1}) + u_t) \\ L(\rho(\cdots \rho(\beta(\rho(u_{t-\tau})) + u_{t-\tau+1})\cdots + u_{t-1}) + u_t) \end{bmatrix},$$

which consists of (80a) and (80b). This implies that the approximation error vector is designed to be $\epsilon_t = 0$.

Since the expression (80a) is the input-output mapping of the true system (77), the true matrix $G^*$ in (8) is $[1\ 0]$. However, we again recall that under adversarial attacks, any estimator cannot distinguish (78) from (77), and may instead recover the input-output mapping of the alternative system (78), resulting in the estimate $\hat{G}_T = [0\ 1]$. This always leads to an estimation error of $\sqrt{2}$.

Now, it remains to calculate $\lambda$ in Assumption 2.4. Let $\gamma$ denote the variable in (80a). Then, we have

$$\mathbb{E}\left[\Phi(\boldsymbol{U}_t^{(\tau)})\Phi(\boldsymbol{U}_t^{(\tau)})^T\right] = \mathbb{E}\left[\mathbb{E}\left[\Phi(\boldsymbol{U}_t^{(\tau)})\Phi(\boldsymbol{U}_t^{(\tau)})^T\right] \mid u_{t-\tau}\right]$$
$$= \mathbb{E}\left[\frac{1}{2}\begin{bmatrix} \gamma^2 & \gamma(\gamma + L\rho^\tau c) \\ \gamma(\gamma + L\rho^\tau c) & (\gamma + L\rho^\tau c)^2 \end{bmatrix} + \frac{1}{2}\begin{bmatrix} \gamma^2 & \gamma(\gamma - L\rho^\tau c) \\ \gamma(\gamma - L\rho^\tau c) & (\gamma - L\rho^\tau c)^2 \end{bmatrix}\right]$$
$$= \mathbb{E}\left[\begin{bmatrix} \gamma^2 & \gamma^2 \\ \gamma^2 & \gamma^2 + (L\rho^\tau c)^2 \end{bmatrix}\right] \tag{81}$$

Note that $\mathbb{E}[\gamma^2] = L\sum_{i=0}^\tau \rho^i$ due to the independence of control inputs and the fact that $\mathbb{E}[u_t^2] = 1$ for all $t$. Let $\mu_{\min}$ denote the minimum eigenvalue of (81). We have

$$\mu_{\min} = \frac{\mathbb{E}[2\gamma^2] + (L\rho^\tau c)^2 - \sqrt{\mathbb{E}[2\gamma^2]^2 + (L\rho^\tau c)^4}}{2} = \frac{\mathbb{E}[2\gamma^2] \cdot (L\rho^\tau c)^2}{\mathbb{E}[2\gamma^2] + (L\rho^\tau c)^2 + \sqrt{\mathbb{E}[2\gamma^2]^2 + (L\rho^\tau c)^4}}$$

$$\geq \frac{2\mathbb{E}[\gamma^2] \cdot (L\rho^\tau c)^2}{\mathbb{E}[2\gamma^2] + (L\rho^\tau c)^2 + \mathbb{E}[2\gamma^2] + (L\rho^\tau c)^2} = \frac{\mathbb{E}[\gamma^2] \cdot (L\rho^\tau c)^2}{\mathbb{E}[2\gamma^2] + (L\rho^\tau c)^2}$$

$$\geq \frac{\mathbb{E}[\gamma^2] \cdot (L\rho^\tau c)^2}{\mathbb{E}[2\gamma^2] + \mathbb{E}[\gamma^2]} = \frac{(L\rho^\tau c)^2}{3},$$

where the first inequality comes from $\mathbb{E}[2\gamma^2] + (L\rho^\tau c)^2 \geq \sqrt{\mathbb{E}[2\gamma^2]^2 + (L\rho^\tau c)^4}$ and the second inequality is due to $\mathbb{E}[\gamma^2] > L\rho^\tau > L\rho^\tau c$. Thus, Assumption 2.4 is satisfied with $\lambda \geq \frac{L\rho^\tau c}{\sqrt{3}}$. In other words, the derived estimation error $\sqrt{2}$ is always lower-bounded by

$$\frac{L\rho^\tau}{\lambda} \cdot \sqrt{\frac{2}{3}} c = \Omega\left(\frac{L\rho^\tau}{\lambda}\right),$$

which completes the proof. □

# F    NUMERICAL EXPERIMENT DETAILS

In this section, we will present experiment details on Section 4. Apple M1 Chip with 8-Core CPU is sufficient for the experiments. The error bars (shaded area) in all the figures in the paper report 95% confidence intervals based on the standard error. We calculate the standard error by running 10 different experiments by generating 10 random sets of matrices $A, B, C$, and $D$ and using random adversarial disturbances for each experiment.

We use the following parameters for the system (20): the state dimension $n = 100$, the control input dimension $m = 5$, the observation dimension $r = 10$, and the time horizon $T = 500$. For the function $\sigma$ that defines $f(x_t, u_t, w_t) = \sigma(Ax_t + Bu_t + w_t)$, we run the experiments with two different $\sigma$:

$$\sigma(x) = \tanh(x) \quad \text{or} \quad \sigma(x) = \text{sgn}(x) \cdot \log(|x| + 1). \tag{82}$$

Both functions are symmetric around the origin, monotonic, and 1-Lipschitz, which are desirable for activation functions of a neural net. Note that the first function is bounded within $[-1, 1]$, while the second function is unbounded. We analyze both options to determine whether the boundedness affects the behavior of the estimation error.

Based on random matrices $A \in \mathbb{R}^{100 \times 100}$, $B \in \mathbb{R}^{100 \times 5}$, $C \in \mathbb{R}^{10 \times 100}$, and $D \in \mathbb{R}^{10 \times 5}$ for each experiment, we build the true input-output mapping for different $\sigma$ options and approximate the mapping to be a linear combination of basis functions as:

$$\sigma(C\sigma(A\sigma(\cdots \sigma(A\sigma(Bu_{t-\tau}) + Bu_{t-\tau+1})\cdots) + Bu_{t-1}) + Du_t) = G^* \cdot \Phi(U_t^{(\tau)}). \tag{83}$$

To this end, we use kernel regression to estimate the true $G^*$ and construct the kernels (basis functions) $\Phi$. The number of kernels used as basis functions is set to $M = 25$. We leverage polynomial kernels of degree up to 3, and select the regularization parameter from $[0.0001, 0.001, 0.01, 0.1, 1, 10, 100]$ based on the one that minimizes the test mean-squared-error. The training and test datasets, split in an 80:20 ratio, are randomly generated from the control inputs whose entries are $\text{Unif}[-15, 15]$ and the corresponding function values based on the left-hand side of (83).

Starting from the initial state $x_0 = 100\mathbf{1}_{100}$, we generate the observation trajectory $y_0, \ldots, y_{T-1}$. Here, $\mathbf{1}_{(\cdot)}$ is the vector of ones with a relevant dimension. Defining $x_t^i$ as the $i^{\text{th}}$ coordinate of $x_t$, when the system is under attack, the adversary selects each coordinate $w_t^i$ of the disturbance $w_t$ to be $\text{sgn}(x_t^i) \cdot \gamma$, where $\gamma \sim N(300, 25)$ if $x_t^i \geq 0$, and $\gamma \sim N(1000, 25)$ otherwise. The control inputs $u_t$ are selected as either one of the following:

$$u_t \sim N(0, 100I_5) \quad \text{or} \quad u_t \sim \text{Unif}[-8, 10]^5. \tag{84}$$

The first is standard zero-mean Gaussian inputs, and the second is nonzero-mean non-Gaussian inputs. We show that both inputs work properly in our setting, in contrast to prior literature that requires zero-mean Gaussian inputs (see Table 1).

The observation trajectory $y_0, \ldots, y_{T-1}$ generated by (20) depends on the hyperparameters $\tau$ and $\rho$. The input memory length $\tau$ affects not only the complexity of (83) but also the attack probability $p$ at each time. We set $p = \frac{1}{2\tau+1}$, which satisfies Assumption 2.8. Moreover, note that $\rho$ is generated by

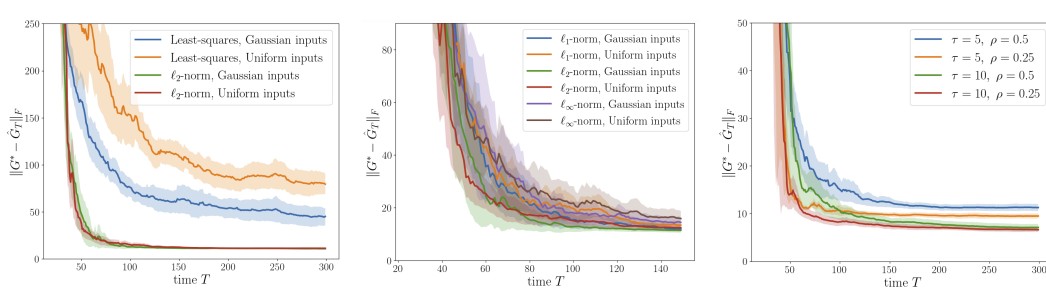

(a) Comparison of Least-squares and $\ell_2$-norm estimator

(b) Comparison of the $\ell_\alpha$-norm estimators ($\alpha = 1, 2, \infty$)

(c) Analysis of $\tau$ and $\rho$ effects under non-Gaussian inputs

Figure 3: Estimation error of the input-output mapping (21) under adversarial attacks under the activation function $\mathrm{sgn}(x) \cdot \log(|x| + 1)$.

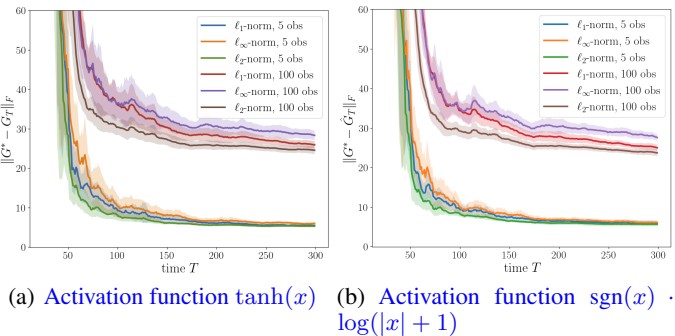

(a) Activation function $\tanh(x)$

(b) Activation function $\mathrm{sgn}(x) \cdot \log(|x| + 1)$

Figure 4: Estimation error of the input-output mapping (21) under adversarial attacks with the two activation functions to analyze how the observation dimension $r$ affects $\ell_\alpha$-norm estimators ($\alpha = 1, 2, \infty$).

adjusting the spectral radius of the matrix $A$. Since both $\sigma$ are 1-Lipschitz functions, $\rho$ in Assumption 2.1 coincides with the spectral radius of $A$ (see Remark 2.2).

Our first experiment compares the $\ell_2$-norm estimator with the commonly used least-squares under $\tau = 5$ and $\rho = 0.5$. We consider both cases of control inputs given in (84). Based on the observation trajectory, we evaluate the following two estimators using the MOSEK solver (MOSEK ApS, 2025):

$$\arg\min_{G} \sum_{t=\tau}^{T-1} \left\| y_t - G \cdot \Phi(\boldsymbol{U}_t^{(\tau)}) \right\|_2 \quad \text{vs.} \quad \arg\min_{G} \sum_{t=\tau}^{T-1} \left\| y_t - G \cdot \Phi(\boldsymbol{U}_t^{(\tau)}) \right\|_2^2 .$$

Our second experiment additionally compares the $\ell_2$-norm estimator with the $\ell_1$-norm estimator and the $\ell_\infty$-norm estimator:

$$\arg\min_{G} \sum_{t=\tau}^{T-1} \left\| y_t - G \cdot \Phi(\boldsymbol{U}_t^{(\tau)}) \right\|_1 \quad \text{and} \quad \arg\min_{G} \sum_{t=\tau}^{T-1} \left\| y_t - G \cdot \Phi(\boldsymbol{U}_t^{(\tau)}) \right\|_\infty .$$

Our third experiment analyzes the effect of $\tau$ and $\rho$ on the $\ell_2$-norm estimator under nonzero-mean uniform inputs, where we consider $\tau \in [5, 10]$ and $\rho \in [0.25, 0.5]$. Note that all experiments were conducted for both activation functions given in (82).

The experiments using $\sigma(x) = \tanh(x)$ are shown in Figure 2 and those with $\sigma(x) = \mathrm{sgn}(x) \cdot \log(|x| + 1)$ are presented in Figure 3. As noted earlier, the two functions differ in their boundedness. In both Figures 2(a) and 3(a), one can observe that the $\ell_2$-norm estimator accommodates both Gaussian and uniform inputs and arrive at a similar stable region, unlike the least-squares estimator.

Furthermore, both Figures 2(b) and 3(b) demonstrate that all norm estimators accurately recover the true matrix $G^*$, with the $\ell_2$-norm estimator achieving the smallest error among them. This supports

the findings in Remark 3.4, which states that only the $\ell_2$-norm estimator attains the optimal error (14) that matches the lower bound presented in Theorem 3.5.

The discrepancy in the estimation error with respect to $\tau$ and $\rho$ can clearly be observed in Figures 2(c) and 3(c), where the estimation error using the $\ell_2$-norm estimator decreases as $\tau$ increases and $\rho$ decreases, which is consistent with our theoretical optimal error of $O(\rho^\tau)$. These findings remain valid regardless of the boundedness of the activation function $\sigma$.

The last experiment given in Figure 4 shows how the observation dimension $r$ may affect the performance of the estimators. In addition to the second experiment comparing the $\ell_2$-norm estimator with the $\ell_1$-norm and $\ell_\infty$-norm estimators, we also test the effect of $r$. To be specific, we compare the partially observed case with the fully observed case; using the observation dimension $r = 5$ or 100. Note that $r = 5$ represents the partially observed case, whereas $r = 100$ represents the fully observed case since $r = n$. Accordingly, the dimension of relevant matrices will be $C \in \mathbb{R}^{r \times 100}$ and $D \in \mathbb{R}^{r \times 5}$.

The results presented in both Figures 4(a) and 4(b) show that increasing $r$ leads to higher estimation error for all three estimators ($\ell_2$, $\ell_1$, and $\ell_\infty$), which agrees with the theoretical results for the $\ell_1$-norm and $\ell_\infty$-norm estimators, while the trend for the $\ell_2$-norm is somewhat milder (see Remark 3.4). Although not perfectly consistent with the analysis that the $\ell_2$-norm estimator may not suffer from increasing $r$, the $\ell_2$-norm estimator remains the least susceptible among the three and continues to achieve the lowest estimation error for a large value of $r$—fully observed case.

## G  NUMERICAL EXPERIMENT ON POWER SYSTEMS

The core assumption of this paper is Assumption 2.8, which states that attack times are sparse, yet the adversary can exploit the full information history at each attack instance. In this section, we illustrate how our setting applies to the real-world systems, and aim to identify the input-output mapping of nonlinear swing dynamics in power grids to show how control inputs to the power system (*i.e.*, mechanical power injections to each node) influence outputs in the presence of adversarial attacks, given only partial observations (*i.e.*, frequencies and rotor angles measured at a limited number of nodes). Below are the symbols related to the power grid.

| Symbol | Description |
|---|---|
| $M_i$ | Inertia constant |
| $D_i$ | Damping coefficient |
| $|E_i|$ | Magnitude of the internal voltage of the generator $i$ |
| $B_{ij}$ | Susceptance between nodes $i, j$ |
| $u_i$ | Mechanical power injection to generator $i$ |
| $w_i$ | Adversarial attack applied to generator $i$ |
| $\delta_i$ | Rotor angle of generator $i$ |
| $\dot{\delta}_i$ | Rotor speed of generator $i$ |

In the power grid applications, consider nonlinear swing dynamics consisting of $n$ different generators:

$$M_i \ddot{\delta}_i + D_i \dot{\delta}_i = h(u_i, w_i) - \sum_{j=1}^{n} |E_i||E_j|B_{ij}\sin(\delta_i - \delta_j), \quad i = 1, \ldots, n, \tag{85}$$

where $M_i, D_i, |E_i|, B_{ij}, G_{ij}$ are unknown parameters of generators, and $\delta_i, \dot{\delta}_i$ are states (generator's rotor angle and rotor speed), $u_i$ is the control input (mechanical power injection to each generator) at node $i$, and $w_i$ is the disturbance applied to each node $i$. The dynamics imply how $i$th generator is affecting and being affected by $j$th generator, for all $i, j = 1, \ldots, n$.

Given the dynamics (85), the goal of the control is to settle every generator's rotor speed $\dot{\delta}_i$ to the nominal grid frequency (e.g. 60Hz in the US); synchronization to turn a collection of individual rotating machines into a single, coherent power-delivery system. Each machine in the grid is dynamically coupled to every other machine, since when you speed up one generator, then the extra power is also applied to every other generator, and then they shift their angles to propagate that power into the rest of the loop, aiming to settle to new angles and back to 60Hz.

Most of the time, the system remains robust, but if undetected disturbances slip through, the system may become destabilized. To this end, we need system identification in the presence of infrequent attacks, where each attack can have an extremely large magnitude. In particular, we need to find the input-output mapping regarding all generators $i = 1, \ldots, n$. The goal is to identify the input-output mapping of the power grid even in the presence of adversarial attacks, where an adversary gains access to the mechanical power injection channel and can inject a malicious perturbation. When a stealthy attack is injected, it will propagate across all grids, causing growing oscillation.

We approximate the dynamics 85 to a discrete-time dynamics with approximating $\sin(\delta_i - \delta_j) \approx \delta_i - \delta_j$. With a sampling time of $t_s = 0.001$, the approximation of $\sin$ and $\cos$ functions are justified. For time $t$, we can write

$$\delta_i(t+1) = \delta_i(t) + t_s \cdot \nu_i(t),$$

$$\nu_i(t+1) = \nu_i(t) + t_s \cdot \frac{1}{M_i} \left[ h(u_i(t), w_i(t)) - \sum_{j=1}^{n} |E_i||E_j| B_{ij}(\delta_i(t) - \delta_j(t)) - D_i \nu_i(t) \right],$$

where $\nu_i = \dot{\delta}_i$.

For parameter values, we set $M_i \in [2, 9], D_i \in [0.2, 1.8], |E_i| \in [0.95, 1.10], B_{ij} \in [5, 15] (B_{ij} = B_{ji})$ for $n$ different nodes.

The relevant system is then

$$\begin{bmatrix} \delta_1(t+1) \\ \vdots \\ \delta_n(t+1) \\ \nu_1(t+1) \\ \vdots \\ \nu_n(t+1) \end{bmatrix} = \begin{bmatrix} I & t_s I \\ K & I - t_s M^{-1} D \end{bmatrix} \begin{bmatrix} \delta_1(t) \\ \vdots \\ \delta_n(t) \\ \nu_1(t) \\ \vdots \\ \nu_n(t) \end{bmatrix} + H(t), \tag{86}$$

where $M = \mathrm{diag}(M_1, \ldots, M_n)$, $D = \mathrm{diag}(D_1, \ldots, D_n)$, and $K \in \mathbb{R}^{n \times n}$ has entries of $K_{ii} = -\frac{t_s |E_i|}{M_i} \sum_{\substack{j=1, \\ j \neq i}}^{n} |E_j| B_{ij}$ and $K_{ij} = \frac{t_s}{M_i} |E_i||E_j| B_{ij}$ for $i \neq j$. Finally, $H(t) = [0, \cdots, 0, \frac{t_s}{M_1} h(u_1(t), w_1(t)), \cdots, \frac{t_s}{M_n} h(u_n(t), w_n(t))]^T$.

Alternatively define $\tilde{H}(t) = [0, \cdots, 0, \frac{0.01}{M_1} h(u_1(t), 0), \cdots, \frac{0.01}{M_n} h(u_n(t), 0)]^T$, and let $A = \begin{bmatrix} I & 0.01 I \\ B & I - \frac{M^{-1} D}{100} \end{bmatrix}$. Recursively applying the system (86) leads to

$$\begin{bmatrix} \delta_1(t) \\ \vdots \\ \delta_n(t) \\ \nu_1(t) \\ \vdots \\ \nu_n(t) \end{bmatrix} = [I \; A \; A^2 \; \cdots \; A^{\tau-1}] \begin{bmatrix} H(t-1) \\ \vdots \\ H(t-\tau) \end{bmatrix} + A^\tau \begin{bmatrix} \delta_1(t-\tau) \\ \vdots \\ \delta_n(t-\tau) \\ \nu_1(t-\tau) \\ \vdots \\ \nu_n(t-\tau) \end{bmatrix}$$

We assume that we can only observe first $r < n$ generators; i.e. $\delta_1, \cdots \delta_r, \nu_1, \ldots, \nu_r$. Thus, the goal is to retrieve 1st to $r$th row, $(n+1)$th to $(n+r)$th row, given $\tilde{H}(t-1), \cdots, \tilde{H}(t-\tau)$, where each $\tilde{H}$ is a unattacked version of $H$ and we know the structure of $h(\cdot, 0)$. The $\ell_2$-norm estimator finds

$$\min_{G \in \mathbb{R}^{2r \times 2n\tau}} \sum_{t=\tau}^{T+\tau-1} \left\| \begin{bmatrix} \delta_1(t) \\ \vdots \\ \delta_r(t) \\ \nu_1(t) \\ \vdots \\ \nu_r(t) \end{bmatrix} - G \begin{bmatrix} \tilde{H}(t-1) \\ \vdots \\ \tilde{H}(t-\tau) \end{bmatrix} \right\|_2.$$

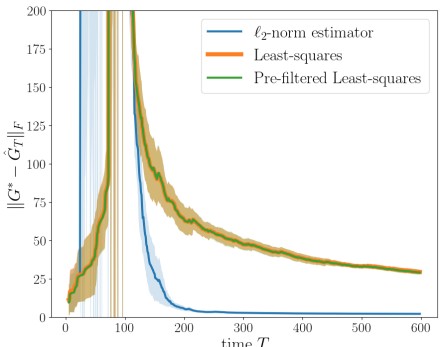

Figure 5: Estimation error of the input-output mapping of nonlinear swing dynamics (85) under adversarial attacks. We compare the performance of the $\ell_2$-norm estimator, ordinary least-squares method, and the pre-filtered least-squares method.

We design the attack vector $w_i(t)$ to leverage the information of the control input $u_i(t)$. At attack times, the adversary selects

$$w_i(t) = \frac{100}{1 + e^{-10000u_i(t)}} |\sin(100t - 200)|,$$

which yields a large positive value when $u_i(t) > 0$, and a large negative value otherwise.

To identify the input-output mapping $[I\ A\ A^2\ \cdots\ A^{\tau-1}]$, we now compare three methods: the $\ell_2$-norm estimator, ordinary least-squares (OLS), and the pre-filtered least-squares method. The $\ell_2$-norm estimator and OLS can respectively be written as

$$\underset{G \in \mathbb{R}^{2r \times 2n\tau}}{\arg\min} \sum_{t=\tau}^{T+\tau-1} \left\| \begin{bmatrix} \delta_1(t) \\ \vdots \\ \delta_r(t) \\ \nu_1(t) \\ \vdots \\ \nu_r(t) \end{bmatrix} - G \begin{bmatrix} \tilde{H}(t-1) \\ \vdots \\ \tilde{H}(t-\tau) \end{bmatrix} \right\|_2, \quad \underset{G \in \mathbb{R}^{2r \times 2n\tau}}{\arg\min} \sum_{t=\tau}^{T+\tau-1} \left\| \begin{bmatrix} \delta_1(t) \\ \vdots \\ \delta_r(t) \\ \nu_1(t) \\ \vdots \\ \nu_r(t) \end{bmatrix} - G \begin{bmatrix} \tilde{H}(t-1) \\ \vdots \\ \tilde{H}(t-\tau) \end{bmatrix} \right\|_2^2.$$

. We also test the performance of a simple version of the pre-filtered least-squares proposed in Simchowitz et al. (2019) can be written as the two-stage least-squares:

$$\hat{K} = \underset{K \in \mathbb{R}^{2r \times 2r}}{\arg\min} \sum_{t=\tau}^{T+\tau-1} \left\| \begin{bmatrix} \delta_1(t) \\ \vdots \\ \delta_r(t) \\ \nu_1(t) \\ \vdots \\ \nu_r(t) \end{bmatrix} - K \begin{bmatrix} \delta_1(t-\tau) \\ \vdots \\ \delta_r(t-\tau) \\ \nu_1(t-\tau) \\ \vdots \\ \nu_r(t-\tau) \end{bmatrix} \right\|_2^2 + \|K\|_F^2$$

$$\implies \underset{G \in \mathbb{R}^{2r \times 2n\tau}}{\arg\min} \sum_{t=\tau}^{T+\tau-1} \left\| \begin{bmatrix} \delta_1(t) \\ \vdots \\ \delta_r(t) \\ \nu_1(t) \\ \vdots \\ \nu_r(t) \end{bmatrix} - \hat{K} \begin{bmatrix} \delta_1(t-\tau) \\ \vdots \\ \delta_r(t-\tau) \\ \nu_1(t-\tau) \\ \vdots \\ \nu_r(t-\tau) \end{bmatrix} - G \begin{bmatrix} \tilde{H}(t-1) \\ \vdots \\ \tilde{H}(t-\tau) \end{bmatrix} \right\|_2^2.$$

Figure 5 demonstrates that the $\ell_2$-norm estimator outperforms the other estimators, which supports the main theme of our paper. In contrast, ordinary least squares is designed to handle only i.i.d.

zero-mean disturbances, and thus its poor performance is expected. Moreover, the pre-filtered least squares method requires the disturbance $w_t$ to be $\mathcal{F}_{t-\tau}$-adapted (see Eq. (1.1) in Simchowitz et al. (2019)), meaning that each disturbance cannot depend on the most recent $\tau$ steps of information history, which is indeed not completely adversarial. In our experiments, attacks were designed to use the most recent inputs, violating this requirement. This explains why the $\ell_2$-norm estimator performs well under Assumption 2.8: sparse attack times, yet each attack is $\mathcal{F}_t$-adapted at attack times $t$.

