# OpenReview forum: "On the Sharp Input-Output Analysis of Nonlinear Systems under Adversarial Attacks"
_ICLR.cc/2026/Conference — Submitted to ICLR 2026_

### Official Review · Reviewer_7vZm · 2025-10-26

**Soundness:** 4
**Presentation:** 2
**Contribution:** 2
**Rating:** 4
**Confidence:** 4

**Summary:**

The paper studies identification of nonlinear dynamical systems with partial observability and potentially adversarial disturbances. It models the system via a finite-memory nonlinear operator G* and analyzes an l2-norm estimator under bounded contraction. The authors prove matching upper and lower bounds on estimation error as functions of the contraction factor and memory length, and show empirical scaling trends on synthetic data. While the analysis is mathematically sound, the motivation and presentation are weak. The adversarial framing is poorly defined, as disturbances are modeled as small-probability corruptions, which is closer to sparse-noise robustness than genuine adversarial robustness.

**Strengths:**

Provides matching upper and lower bounds for a finite-memory setting. Handles non-Gaussian and correlated disturbances beyond the standard i.i.d. noise model. The mathematical exposition of the main theorems is careful and logically consistent.

**Weaknesses:**

The paper’s motivation and exposition are weak. The “adversarial attack” model corresponds only to rare sample corruptions and is not meaningfully adversarial, and it is not explained what real scenarios are captured by this model. While the authors argue that the small-probability corruption assumption is needed due to existing lower bounds, there are alternative approaches to handling this issue. In particular, rather than focusing on probabilistic sparsity assumptions, one could aim to characterize the fundamental learnability of nonlinear dynamical systems directly. Some examples in that direction (not cited in the paper) are: Safely Learning Dynamical Systems (Ahmadi, Chaudhry, Sindhwani & Tu, 2024) and Efficient PAC Learnability of Dynamical Systems Over Multilayer Networks (Qiu, Adiga, Marathe, Ravi, Rosenkrantz, Stearns & Vullikanti, 2024). A very recent example to an attempt in that direction is Universal Learning of Nonlinear Dynamics (Dogariu, Brahmbhatt & Hazan, 2025), and the present work could benefit from discussing the conceptual relationship. All of that to say, it is very unclear why is it relevant to study small probability disturbances in the context of controlling unknown nonlinear dynamical systems.

More broadly, the paper lacks engagement with the substantial machine-learning view of control literature, which is particularly expected at an ML venue. For example, Learning Nonlinear Dynamical Systems from a Single Trajectory (Foster, Rakhlin & Sarkar, 2020) already provides Lyapunov-based conditioning and optimal sample-complexity guarantees for learning a two-layer ReLU dynamical model. A review of more works on learning linear dynamical systems is missing, for example No-Regret Prediction in Marginally Stable Systems (Ghai, Lee, Singh, Zhang & Zhang, 2020) and A New Approach to Learning Linear Dynamical Systems (Bakshi, Liu, Moitra & Yau, 2023). In addition, Improper Learning for Non-Stochastic Control (Simchowitz, Singh & Hazan, 2020) tackles control of unknown linear dynamical systems under adversarial disturbances. The paper does not cite or connect to these lines of work, which undermines its relevance to the ML and online control communities.

Finally, while the theoretical contribution is sound, it is fairly limited in relevance, and the experiments are preliminary: small synthetic systems only, no real data, no ablations, and no comparison to prior algorithms.

**Questions:**

What realistic scenario motivates the “adversarial disturbance” model beyond occasional corruptions?

Why is it relevant to extend the input structure (non-gaussian)? Isn't it chosen by the learner? On that note, how is this method more general than considering the system without inputs?

---

> ### Author Response · Authors · 2025-11-20
>
> We are very grateful to the reviewer for a thorough evaluation and providing valuable feedback. We improved the paper based on the provided comments. Please see the *blue-highlighted* parts for the updated draft.
>
> ## 1. Small-probability corruption assumption and Realistic scenarios
> We thank the reviewer for raising the important concern about *Assumption 2.8* and providing a bunch of literature. We would like to respond from two perspectives: theoretical motivation and potential realistic scenarios.
>
> > ### Theoretical Motivation
>
> - Consider the case where the *adversary is fully informed*, meaning that the adversary **fully leverages the information history** $\mathcal{F}_t = \\{x_0, \dots, x_t\\}$ to generate the attack $w_t$ **at every time $t$**. Then, the outputs will be entirely corrupted at every time step, and it will be impossible to achieve an accurate estimation. To illustrate, the recent result in [1] (which the reviewer had kindly mentioned) shows that the constant error term $\bar w$ cannot be eliminated when only the bound $\\|w_t\\|_2 \le \bar w$ is known for all $t$ (see their Thm 3.3(c)). What's worse, in our setting where each attack is sub-Gaussian (not necessarily bounded), we would have $\bar w=\infty$, leading to the error being completely meaningless. Therefore, additional constraints on the adversary are necessary—such as *"limiting the adversary’s information"* or *"limiting the number of attack times"*—to make accurate system identification possible.
>
> - A good example of the former approach (**limiting the adversary's information**) is the work [2],[3] (you kindly mentioned [3], which builds upon the algorithms in [2]). To illustrate, the underlying assumption on those works is that the adversary is oblivious and ends up injecting semi-adversarial disturbances in the sense that the disturbance at time $t$ is $\mathcal{F}_0$-adapted (see Assumption 6b in [3]). That being said, the adversary cannot consider the livestream information history to design the attack.
>
> - Now, our paper takes the latter approach (**limiting the number of attack times**). In other words, our goal is to identify the system when the attack times are sparse. Instead, note that we now allow that the adversary can design attacks based on the full information history $\mathcal{F}_t$ when $t$ is an attack time. Which aspect of the adversary we restrict—information or attack times—depends on the context, and we will later explain that our setting also has strong applicability.
>
> - Still, we fully acknowledge the reviewer's concern that a more general scenario would be to consider the composite noise that consists of both adversarial part and stochastic part. However, even without the stochastic noise and thus many attacks being exactly zero, we find that adversarial attacks can only be overcome if the attack probability $p<O(1/\tau)$;  otherwise, adversarial attacks can completely destroy the system. Our analysis provides a sharp characterization, showing that $O(\rho^\tau)$ is an optimal error when the adversary has access to the full information history, even with sparse attack times. Sharpness here refers to Thm 3.1 as the upper bound and Thm 3.4 as the lower bound.
>
> - We have added a brief comparison between [2] and our work in Lines 607-613. While this discussion currently appears in the appendix, we will move it to the main content of the camera-ready version should the paper be accepted.
>
> > ### Potential Realistic Scenarios
>
> - We would like to justify our sparse attack times assumption through the most important application domains: cyberphysical systems [4]. Cyberphysical systems such as power grids, transport networks, or autonomous vehicles, are already equipped with advanced security mechanisms. When cyberattacks are injected into the system, they can broadly be classified as detectable or undetectable. In practice, most attacks are detectable by well-designed detectors and are effectively nullified or corrected by well-built controllers, and thus the system dynamics are not corrupted (this is the case where the attack is exactly zero).
>
> - However, undetectable attacks can occur when a strong adversary leverages *full information history* to craft a sophisticated and malicious attack [5]. These so-called *stealthy* attacks slips through the system while remain unnoticed. For cyberphysical systems, these attacks are indeed possible but happens infrequently [6]. For example, our model completely captures the most realistic scenario for **cyberattacks on power systems**. In that application, an attacker stealthily changes the communication between different nodes in the system, which makes the operator inject a wrong amount of mechanical power into the system at certain times. This will make the system deviate from equilibrium and cause a blackout. We believe that this is well-explained in Lines 88-94, but we now included in the draft that this indeed applies to cyberphysical systems.
>
> (Continued)

---

> > ### Author Response · Authors · 2025-11-20
> >
> > ## 2. Real-World Experiments on Power Systems
> >
> > - Regarding the discussion on power systems in the above rebuttal, we conducted experiments and included them in the revised draft (see Lines 471-473 for a pointer and Pages 27-30 for details). To summarize, The goal is to identify the input-output mapping of the nonlinear swing dynamics in a power grid with $n$ generators, where an adversary can occasionally apply arbitrary power injections:  $$M_i \ddot \delta_i + D_i \dot \delta_i= h(u_i,w_i) - \sum_{j=1}^n |E_i||E_j|B_{ij} \sin(\delta_i - \delta_j), \quad i=1,\dots, n, $$
> > where $u_i$ is the control input (mechanical power injection to each generator) at node $i$, $w_i$ is the disturbance applied to each node $i$, and the rest symbols are constants. Please see the remaining details in Pages 27-30. Currently, these results are in the appendix, but we will be sure to include them in the main content of the camera-ready version.
> >
> > - We tested three estimators to this power grid identification problem. First, our proposed $\ell_2$-norm estimator. Second, ordinary least squares, a well-studied estimator in the literature. Third, the pre-filtered least-squares method proposed in [2]. Since the assumptions in [2] and ours represent *two different ways of limiting the adversary's power* as discussed, it is interesting to investigate whether the third method fails under our setting. In fact, the $\ell_2$-norm estimator outperforms the other estimators, which supports the main theme of our paper (see Figure 5).
> >
> > - In this paper, we focus on obtaining the $\ell_2$-norm estimate of the true mapping $G^\ast$. Therefore, it is fair to compare our approach with other point estimators. Most existing methods are based on ordinary least squares, and the estimator in [2] is essentially a variant of least squares. The experiments compare these methods and demonstrate how the non-smooth $\ell_2$-norm estimator outperforms smooth least-square-based estimators.
> >
> > ## 3. Discussion on Control Literature
> >
> > - You kindly suggested several additional literature other than [1],[3], and we would like to clarify why we did not directly connect those works with ours.
> >
> > - The works [7],[8] focus primarily on the learnability of dynamical systems and provide a framework to determine whether a system can be learned in principle. While we appreciate these contributions, they do not directly study the impact of adversarial disturbances. In contrast, our objective is more targeted: we aim to obtain an accurate point estimate of the true $G^\ast$ under adversarial attacks.
> >
> > - The works [9],[10] consider the setting in which disturbances are i.i.d. and zero-mean (not adversarial), and are not directly relevant to our paper, which focuses on adversarial disturbances. Nonetheless, we believe that [10] is a good literature for an alternative approach to linear systems, so we now included this explanation in Lines 605-607. Thanks for providing the literature!
> >
> > - The work [11] studies bounded adversarial disturbances. When designing a controller for unknown system, the system identification should precede, and most literature (including [11]) takes *least-squares-type algorithm*. In particular, [11] used least-squares + regularization term. Theorem 3 in [11] guarantees good performance for some regularization constant, but its value is completely unknown and impossible to retrieve under adversarial disturbances. This is why [11] **could not include any numerical experiments**, while our work provides a fair amount of experiments. This highlights the need to limit the adversary, for which our work restricted the number of attack times.
> >
> > ## 4. Discussion on Input Structure
> >
> > - In our Remark 2.7, we discussed why it is important to consider non-Gaussian/nonzero-mean control inputs. We fully agree with the reviewer that, since the control inputs are chosen by the learner, the system identification procedure itself is barely affected. However, the learner may want to choose non-Gaussian, nonzero-mean inputs to **improve performance** (e.g., minimize costs), which can serve as a *secondary benefit during system identification*. In feedback control, it is common to select inputs of the form “$K(y_t)$ + [excitation term]”, where $K(y_t)$ may not have zero-mean and [excitation term] may not be Gaussian, showing that good performance can be pursued while concurrently maintaining accurate system identification.
> >
> > - Based on your comment, we clarified Remark 2.7 and added more details about this secondary benefit in Lines 244-247.
> >
> > ### We hope that our rebuttal helps clarify our work. Please let us know any further concerns or questions. Many thanks!

---

> > > ### Author Response · Authors · 2025-11-20
> > >
> > > References
> > >
> > > [1] Dogariu et al., "Universal Learning of Nonlinear Dynamics", arXiv 2025.
> > >
> > > [2] Simchowitz et al., "Learning Linear Dynamical Systems with Semi-Parametric Least Squares", COLT 2019.
> > >
> > > [3] Simchowitz et al., "Improper Learning for Non-Stochastic Control", COLT 2020.
> > >
> > > [4] Pasqualetti et al., "Attack Detection and Identification in Cyber-Physical Systems", IEEE TAC, 2013.
> > >
> > > [5] Mo and Sinopoli, "Secure Estimation in the Presence of Integrity Attacks", IEEE TAC, 2015.
> > >
> > > [6] Giraldo et al., "A Survey of Physics-Based Attack Detection in Cyber-Physical Systems", ACM Computing Surveys, 2018.
> > >
> > > [7] Ahmadi et al., "Safely Learning Dynamical Systems", FoCM, 2025.
> > >
> > > [8] Qiu et al., "Efficient PAC Learnability of Dynamical Systems Over Multilayer Networks", ICML 2024.
> > >
> > > [9] Foster et al., "Learning Nonlinear Dynamical Systems from a Single Trajectory", L4DC 2020.
> > >
> > > [10] Bakshi et al., "A New Approach to Learning Linear Dynamical Systems", STOC 2023.
> > >
> > > [11] Ghai et al., "No-Regret Prediction in Marginally Stable Systems", COLT 2020.
> > >
> > > Best,
> > >
> > > Authors

---

> ### Author Response · Authors · 2025-11-27
> **Kind Request for Feedback**
>
> Dear Reviewer 7vZm,
>
> As the discussion period will end in a few days, we would greatly appreciate it if you could review our responses and let us know if there are any further concerns or issues that we can address. Please also take a look at our revised draft, which improves the paper based on your comments and includes additional real-world power-system experiments on pages 27–30. We hope that you will kindly reconsider your score if our responses are satisfactory. Many thanks.
>
> Best,
>
> Authors

---

### Official Review · Reviewer_c6MH · 2025-10-29

**Soundness:** 3
**Presentation:** 2
**Contribution:** 3
**Rating:** 6
**Confidence:** 3

**Summary:**

In this paper, researchers study the problem of learning input-output mapping for general nonlinear dynamical systems under disturbances from adversarial attacks. The paper’s work has focused on nonlinear systems with partially observed outputs, non-Gaussian control inputs, and correlated, nonzero-mean, possibly adversarial disturbances. This setting significantly expands the scope of the existing literature and significantly broadens the range of allowed control inputs. This paper reformulates a nonlinear dynamical system as a linear combination of basis functions and proves that l2-norm estimator overcomes the challenge as long as the probability of adversarial attacks on the system at a given moment is less than a certain threshold. This paper presents an estimation error bound that decays with the length of the input memory, and proves the optimality of the estimation error bound by constructing a problem instance.

**Strengths:**

S1: The paper relax the range of control inputs for the task of identifying nonlinear dynamic systems, and require only partial observed outputs, allowing adversarial disturbances with nonzero-mean.

S2: The related work of this paper is fully investigated and the research gap in related fields is accurately grasped. Meanwhile, researchers propose the appropriate problem formulation and theoretical analysis.

S3: The paper has a clear framework and the writing is fluent. The problem positioning and the theoretical derivation steps are clear.

S4: The setting of input and disturbance in this paper is closer to the real scene, which is more practical than previous studies.

**Weaknesses:**

W1: The outline of your analysis is clear, but some parts lack specific details. Moving some of the reasoning from the appendix to the main body might make this section more readable.

W2: In addition to proving that the l2 norm estimation is optimal, the experiment should also compare the effect of partial observation output and full observation output on the error.

W3: Some symbols lack interpretation, leading to a reduction in the readability of parts, such as E for Equation (10) and ψ2 for Assumption 2.5.

**Questions:**

Q1: Your point is that your theory applies to general nonlinear dynamical systems. Does this mean that general nonlinear dynamical systems have Lipschitz continuity?

Q2: Are the basis functions used for nonlinear dynamic systems all nonlinear?

Q3: Do sub-Gaussian variables have any special impact on evaluating nonlinear dynamic systems compared to general variables?

---

> ### Author Response · Authors · 2025-11-19
>
> We are very grateful to the reviewer for a thorough evaluation and providing valuable feedback. We improved the paper based on the provided comments. Please see the *blue-highlighted* parts for the updated draft.
>
> > W1 (Details on analysis)
>
> - Thank you for your thoughtful comment! Unfortunately, we are currently unable to include additional details in the main text due to page limitations. If the paper is accepted, we will expand Section 3 in the camera-ready version. For now, we added Line 361 addressing your Q3 (further explanation follows below) and, if allowed, will include more details on how sub-Gaussian variables along with Lipschitz continuity may help the analysis.
>
> > W2 (more experiments on partial vs. full)
>
> - We really appreciate this suggestion. As you noted, Remark 3.4 points out that the $\ell_2$-norm estimator outperforms $\ell_1$- or $\ell_\infty$-norm estimators, especially because the latter estimators will be more susceptible to the observation dimension $r$. Figures 2(b) and 3(b) only showed for a fixed $r$ that the $\ell_2$-norm estimator outperforms the others. Thus, Based on the reviewer’s suggestion, we added an experiment analyzing the impact of $r$ on the estimation error, comparing the partial observation case ($r<n$) with the full observation case ($r = n$).
>
> - The results show that increasing $r$ leads to higher estimation error for all three estimators ($\ell_2$, $\ell_1$, and $\ell_\infty$), which agrees with the theoretical results for the $\ell_1$- and $\ell_\infty$-norm estimators, while the trend for the $\ell_2$-norm is somewhat milder (see Remark 3.4). Although not perfectly consistent with the analysis that the $\ell_2$-norm estimator may not suffer from increasing $r$, the $\ell_2$-norm estimator remains the least susceptible among the three and continues to achieve the lowest estimation error for a large value of $r$—fully observed case. The new experiments are included in Appendix F (see Lines 1411-1423, and Figure 4 in pages 26-27) for now, but if the reviewer considers it necessary, we will move this content to the main paper for the camera-ready version.
>
> > W3 (readability)
>
> - $\mathbb{E}$ is the expectation operator. We hope that **Notation.** part in Section 1 helps understand the symbol. We acknowledge that the definition of $\psi_2$ may not be immediately accessible to new readers, but we provide a detailed explanation of sub-Gaussian variables and the $\psi_2$-norm in Appendix B. Line 230 now includes a pointer to Appendix B, directing readers to the full definitions.
>
> > Q1 (General nonlinear systems)
>
> - Our analysis will only be available for Lipschitz continuous dynamical systems, since it should satisfy Assumption 2.1. To clarify the scope of our analysis, we added in Line 101 (**Contribution.** part in Section 1) that we will only consider Lipschitz continuous nonlinear systems. Thanks!
>
> > Q2 (Nonlinear basis functions)
>
> - The choice of basis functions is at the discretion of the analyst. In general, it is preferable to select basis functions that are sufficiently expressive. A common choice is radial basis functions (RBFs), among which the Gaussian RBF takes the form $\exp(-\\|x - c\\|^2 / (2\sigma^2))$ (indeed nonlinear) for some vector $c$ and positive scalar $\sigma$. Function approximation theory states that, with a sufficiently large number of such RBFs, any continuous function can be approximated as a linear combination of these nonlinear basis functions. These details are now included in Lines 216-218.
>
> > Q3 (sub-Gaussian on the analysis)
>
> - Your conjecture is indeed true. Since the disturbances and inputs are sub-Gaussian variables, the states will also be sub-Gaussian under Assumption 2.1 (Lipschitz continuity). See Lemma D.9 for the proof that the states are sub-Gaussian. This allows us to upper-bound the right-hand side of Eq. (17), which scales (due to Lemma 2.3) with
> \\[(\rho^\tau \\|x_{t-\\tau}\\|_2 + \bar \epsilon)\cdot T.\\]
>
> - Therefore, to ensure that the above expression scales with $T$, our analysis requires $\\|x_{t-\\tau}\\|_2$ to be bounded by a finite constant, which is guaranteed by the sub-Gaussian assumption. Without this assumption, the state could explode and a finite bound is unattainable. We added this clarification in Line 361.
>
> We hope that our rebuttal helps clarify our work. Please let us know any further concerns or questions. Many thanks!
>
> Best,
>
> Authors

---

> ### Author Response · Authors · 2025-11-27
> **Kind Request for Feedback**
>
> Dear Reviewer c6MH,
>
> As the discussion period will end in a few days, we would greatly appreciate it if you could review our responses and let us know if there are any further concerns or issues that we can address. Please also take a look at our revised draft, which improves the paper based on your comments and includes additional real-world power-system experiments on pages 27–30. We hope that our responses will help solidify your positive evaluation. Many thanks.
>
> Best,
>
> Authors

---

### Official Review · Reviewer_UBeX · 2025-11-01

**Soundness:** 2
**Presentation:** 2
**Contribution:** 2
**Rating:** 6
**Confidence:** 2

**Summary:**

The paper considers the problem of learning the mapping from the control input to the partial observation of a nonlinear dynamical system under more general assumptions on the control input and the allowed adversarial attacks. The error bound is also presented for the proposed approximation model.

**Strengths:**

Overall, the paper considers an interesting problem and proposes an effective numerical method.

**Weaknesses:**

The presentation of the paper need to be revised based on the comments in the Questions part.

**Questions:**

1. Eq. (1) The problem is not properly stated. What is the meaning of "partially observed"? What is w in the original problem, which would not be the adversarial disturbances. What are f and g?. Are both of them totally unknown? Is x_0 given?

2. G^\ast and \Phi a in Eq. (2) are very unclear and misleading. \Phi is just a set of M real-valued functions defined on (U)^{\tau+1}, and each component of the first term on the RHS of (2) is the linear combination of them. Also, I think the dimension of G^\ast should be r by M. In addition, it is unclear how G^\ast is defined in terms of (2) and (2) in the current form seems to be independent of w.

3. lines 81-86 on page 2. The explanation here is also misleading. To make the approximation error smaller, one would usually need to increase the number of the basis functions.

4. Lines 218-220. I cannot understand the explanation in the bracket.

5. It would be helpful to provide some examples of the basis functions after assumption 2.4. How to choose the basis functions in practice?

6. Again, it is unclear how G^\ast is defined in terms of (7).

7. Theorem 3.1. The paper is aimed at approximating the input-output map. Then the main result should be stated in terms of the output error instead of the error of G and G^\ast.

---

> ### Author Response · Authors · 2025-11-19
>
> We are very grateful to the reviewer for a thorough evaluation and providing valuable feedback. We improved the paper based on the provided comments. Please see the *blue-highlighted* parts for the updated draft.
>
> > Q1 (Proper setup)
>
> - Thanks for the comment. We added further explanations on $f$ and $g$ (Lines 46-47). Specifically, $f$ denotes the system dynamics, which connects between the states $x_t$ and $x_{t+1}$, while $g$ denotes the measurement model which connects between the state $x_t$ and the observation $y_t$.
>
> - *Partially observed system* is a standard term in control theory referring to the setting in which we observe only $y_t$, not the full state $x_t$. For example, if $x_t$ is a 10-dim vector, a partial observation might be $y_t$ containing only the first five entries of $x_t$. We have access only to the data $y_t$'s instead of $x_t$'s.
>
> - We also added that our goal is to identify the input–output mapping induced by $f$ and $g$ only based on the collected data $\\{u_t, y_t\\}_{t=0}^{T-1}$ (Line 49), which means that both $f$ and $g$ are unknown. The $w_t$'s are adversarial disturbances.  $x_0$ is also unknown, but the corresponding $y_0$ is known to us, since it is the observation at time $0$.
>
> > Q2 ($G^\ast, \Phi$ in Eq. (2))
>
> - Your comment "$\Phi$ is just ... of them" is exactly correct. That is precisely the key to our analysis. Please note that the input-output mapping we need to identify is \\[g\circ f \circ \dots \circ f(\cdot),\\] which would be an **unknown** nonlinear function (see Step 2 in Lines 189-196).
>
> - Each basis function in $\Phi$ can be selected at the analyst’s discretion (ideally using sufficiently expressive choices such as radial basis functions), and thus these functions are already **known** to us (see Lines 81–86).
>
> - Then, function approximation theory enables us to reformulate this function as a **unknown** linear combination of **known** basis functions $G^\ast \cdot \Phi(\cdot)$ up to a small approximation error  (see Step 3 in Lines 198-209).
> In this setting, Our goal is to learn the **unknown** coefficient matrix $G^\ast$. Based on your comment, we added a pointer in Line 87 to guide readers.
>
> - Thank you pointing out the typo. The $r\times M$ is indeed correct. We have fixed it.
>
> - Eq. (2) is intentionally written in a schematic form. We noted that the ‘residual terms’ are functions of disturbances and far-past states (Line 78). Based on the reviewer's comment, we revised this part to specify that the relevant disturbances are $w_{t-1}, \dots, w_{t-\tau}$, and the past state refers to $x_{t-\tau}$. We hope this makes it clear that Eq. (2) includes the dependence on the $w_t$'s.
>
> > Q3 (Lines 81-86)
>
> - Your comment is exactly correct. The number of basis functions $M$ can be chosen large enough to ensure that the approximation error is sufficiently small. The analyst is free to select both the number and the structure of the basis functions, and any finite $M$ is acceptable.
>
> - One may question whether the coefficient matrix $G^\ast$ is unique given a desired approximation error, so we have added a clarification: we do not require $G^\ast$ to be uniquely defined; the result in Thm 3.1 holds for any $G^\ast$ that satisfies \\[\\|g\circ f\circ \dots \circ f(\cdot ) -  G^* \Phi(\cdot)\\|_2 \le \bar\epsilon\\] for all points in the domain of interest, given a small approx. error $\bar \epsilon\ge 0$. See revised in Lines 201, 208, 284, 293.
>
> > Q4 (Lines 218-220)
>
> - We agree with the reviewer that the original explanation may have been confusing. The purpose was to note that function approximation theory applies only to *continuous* nonlinear functions. Since our function of interest $g\circ f\circ \dots \circ f$ is Lipschitz continuous (by Assumption 2.1), selecting Lipschitz continuous basis functions would be a natural approach. To make this point clearer, we added a new explanation in Lines 216–218 and removed the original text.
>
> > Q5 (Basis functions)
>
> - We briefly included in Lines 216-218 that polynomials or radial basis functions can be examples of basis functions. The number of basis functions $M$ is chosen freely, as stated in Line 207.
>
> > Q6 ($G^\ast$)
>
> - We hope that the definition of $G^\ast$ is now clear, based on the responses to Q2 and Q3.
>
> > Q7 (input-output mapping)
>
> - Throughout the paper, $G^\ast$ denotes the coefficient matrix that captures the input–output mapping up to a small approx. error. From the true input-output mapping $g\circ f\circ \dots \circ f$, we obtain $G^\ast\cdot \Phi(U_t^{(\tau)})$ (Eq. (2) or (8)), which approximately represents a mapping from $U_t^{(\tau)}$ to $y_t$ in the absence of attacks. Since $\Phi$ is known to us, now the goal is to accurately estimate $G^\ast$. This is why we stated that estimating $G^\ast$ is equivalent to identifying the correct input–output mapping.
>
> We hope that our rebuttal helps clarify our work. Please let us know any further concerns or questions. Many thanks!
>
> Best,
>
> Authors

---

> ### Author Response · Authors · 2025-11-27
> **Kind Request for Feedback**
>
> Dear Reviewer UBeX,
>
> As the discussion period will end in a few days, we would greatly appreciate it if you could review our responses and let us know if there are any further concerns or issues that we can address. Please also take a look at our revised draft, which improves the paper based on your comments and includes additional real-world power-system experiments on pages 27–30. We hope that our responses will help solidify your positive evaluation. Many thanks.
>
> Best,
>
> Authors

---

### Official Review · Reviewer_i3Yb · 2025-11-11

**Soundness:** 3
**Presentation:** 3
**Contribution:** 2
**Rating:** 4
**Confidence:** 3

**Summary:**

This paper proposes to approximate the (finite impute-response functions or) input-output characteristics of a (partially observable) nonlinear system by a linear function in a given basis. The proposed method works in presence of sporadically occurring adversarial disturbances and non-Gaussian control inputs.

**Strengths:**

Indeed, quite a bit of work in this area in either on linear systems, or for IID perturbations. So this addresses a novel facet of the sysid problem.

The final bound is quite interpretable.

**Weaknesses:**

I am fine with most assumptions in the paper, to the extent such assumptions (like spectral radius like condition) are also required in the linear case. However, Assumption 2.8 sticks out like a sore thumb. In general, it is a reasonable expectation that works dealing with adversarial disturbances generalize the stochastic case; for example, this is the case for Simchowitz et al who can handle an oblivious adversary (I can give more example here if needed). But, Assumption 2.8 implies that most of the time the entire truncated history of perturbations is exactly zero. Thus, not only does it not generalize the stochastic case, but also, but it also places limits on identifiably of G even in the infinite sample regime. Now, irrespective of how apt this assumption is, it severely limits the applicability of this work. At the very least, this should be noted early, e.g., in the abstract. (I also wonder if a condition like that in Theorem 3.3 is necessary in this regime, or if something qualitatively weaker that captures the fact that the first few most recent w's are more likely to be uncorrupted qualifies.)

Lastly, in statistical learning, when approximating a function by a class that does not contain it, the best approximation also depends on the distribution of inputs (i.e., holds in a distributional L1 sense). I suspect this also holds here (I didn't see a concrete definition of G*). This would also limit using G* for control purposes (or under different inputs downstream).

**Questions:**

Can the authors please point out any aspect of the work (as summarized above) I might have misunderstood?

---

> ### Author Response · Authors · 2025-11-19
>
> We are very grateful to the reviewer for a thorough evaluation and providing valuable feedback. We improved the paper based on the provided comments. Please see the *blue-highlighted* parts for the updated draft.
>
> ## Q1 (Assumption 2.8)
> We thank the reviewer for raising the important concern about *Assumption 2.8*. We would like to respond from two perspectives: theoretical motivation and potential applications.
>
> > ### Theoretical Motivation
>
> - Consider the case where the *adversary is fully informed*, meaning that the adversary **fully leverages the information history** $\mathcal{F}_t = \\{x_0, \dots, x_t\\}$ to generate the attack $w_t$ **at every time $t$**. Then, the outputs will be entirely corrupted at every time step, and it will be impossible to achieve an accurate estimation. To illustrate, the recent result in [1] shows that the constant error term $\bar w$ cannot be eliminated when only the bound $\\|w_t\\|_2 \le \bar w$ is known for all $t$ (see their Theorem 3.3(c)). What's worse, in our setting where each attack is sub-Gaussian (not necessarily bounded), we would have $\bar w=\infty$, leading to the error being completely meaningless. Therefore, additional constraints on the adversary are necessary—such as *"limiting the adversary’s information"* or *"limiting the number of attack times"*—to make accurate system identification possible.
>
> - A good example of the former approach (**limiting the adversary's information**) is the work [2, 3], which the reviewer has kindly pointed out. To illustrate, the underlying assumption on those works is that the adversary is oblivious and ends up injecting semi-adversarial disturbances in the sense that the disturbance at time $t$ is $\mathcal{F}_{t-T}$-adapted (see Eq. (1.1) in [2]) for a given $T>0$. That being said, the adversary cannot consider the $T$ most recent information history to design the attack. Note that the constant $T$ in [2] is analogous to the constant $\tau$ in our paper (which defines the attack probability $O(1/\tau)$).
>
> - Now, our paper takes the latter approach (**limiting the number of attack times**). In other words, our goal is to identify the system when the attack times are sparse. Instead, note that we now allow that the adversary can design attacks based on the full information history $\mathcal{F}_t$ when $t$ is an attack time. Which aspect of the adversary we restrict—information or attack times—depends on the context, and we will later explain that our setting also has strong applicability.
>
> - Still, we fully acknowledge the reviewer's concern that a more general scenario would be to consider the composite noise that consists of both adversarial part and stochastic part. However, even without the stochastic noise and thus many attacks being exactly zero, we find that adversarial attacks can only be overcome if the attack probability $p<O(1/\tau)$;  otherwise, adversarial attacks can completely destroy the system. Our analysis provides a sharp characterization, showing that $O(\rho^\tau)$ is an optimal error when the adversary has access to the full information history, even with sparse attack times. Sharpness here refers to Thm 3.1 as the upper bound and Thm 3.4 as the lower bound.
>
> - We have added a brief comparison between [2] and our work in Lines 607-613. While this discussion currently appears in the appendix, we will move it to the main content of the camera-ready version should the paper be accepted.
>
> > ### Potential Applications
>
> - We would like to justify our sparse attack times assumption through the most important application domains: cyberphysical systems [4]. Cyberphysical systems such as power grids, transport networks, or autonomous vehicles, are already equipped with advanced security mechanisms. When cyberattacks are injected into the system, they can broadly be classified as detectable or undetectable. In practice, most attacks are detectable by well-designed detectors and are effectively nullified or corrected by well-built controllers, and thus the system dynamics are not corrupted (this is the case where the attack is exactly zero).
>
> - However, undetectable attacks can occur when a strong adversary leverages *full information history* to craft a sophisticated and malicious attack [5]. These so-called *stealthy* attacks slips through the system while remain unnoticed. For cyberphysical systems, these attacks are indeed possible but happens infrequently [6]. For example, our model completely captures the most realistic scenario for **cyberattacks on power systems**. In that application, an attacker stealthily changes the communication between different nodes in the system, which makes the operator inject a wrong amount of mechanical power into the system at certain times. This will make the system deviate from equilibrium and cause a blackout. We believe that this is well-explained in Lines 88-94, but we now included in the draft that this indeed applies to cyberphysical systems.
>
> (Continued)

---

> > ### Author Response · Authors · 2025-11-19
> >
> > - Meanwhile, we agree with the reviewer that our assumptions should be clearly stated in the abstract. Previously, we used the expression "*as long as the probability ... is smaller than a certain threshold*," which implicitly corresponds to $p<O(1/\tau)$. We have now revised this phrasing in the abstract to emphasize that the attack times are sparse, while the adversary has access to the full information history when designing its attacks.
> >
> > ## Q2 (Function Approximation)
> >
> > - We really appreciate this question. We agree that the definition of $G^\ast$ may not be concrete in the original draft. However, we note that the input distribution does not affect our approach. In our work, we assume the use of sufficiently expressive basis functions $\Phi=(\phi_1, \dots, \phi_M)$ that can approximate functions *over the entire domain*. Function approximation theory guarantees that a wide class of nonlinear mappings can be approximated to arbitrary precision (see Lines 81–87). In other words, for any desired approximation tolerance $\bar\epsilon \ge 0$, there exists a coefficient matrix $G^\ast$ such that $$\\|g\circ f\circ \dots \circ f(\cdot ) -  G^\ast \Phi(\cdot)\\|_2 \le \bar\epsilon$$ for *all points in the domain of interest*. We included this *universal approximation* in the revised draft, and hope that Line 87 (pointing to Lines 198-209), Line 201, and Eq. (7) now help clarify any previous confusion.
> >
> > - We note that there can be multiple $G^\ast$ that satisfies Eq. (7) when constructing an approximation; nevertheless, our following theorems hold for any such matrix $G^\ast$; *e.g.*, an upper bound on $\\|G^\ast - \hat G_T\\|_F$ in Thm 3.1 holds for any $G^\ast$ that universally approximates the function within a given tolerance $\bar \epsilon$. We included these details in Lines 208-209, Lines 284-285, and Lines 293-294.
> >
> > ### We hope that our rebuttal helps clarify our work. Please let us know any further concerns or questions. Many thanks!
> >
> > References
> >
> > [1] Dogariu et al., "Universal Learning of Nonlinear Dynamics", arXiv 2025.
> >
> > [2] Simchowitz et al., "Learning Linear Dynamical Systems with Semi-Parametric Least Squares", COLT 2019.
> >
> > [3] Simchowitz et al., "Improper Learning for Non-Stochastic Control", COLT 2020.
> >
> > [4] Pasqualetti et al., "Attack Detection and Identification in Cyber-Physical Systems", IEEE Transactions on Automatic Control, 2013.
> >
> > [5] Mo and Sinopoli, "Secure Estimation in the Presence of Integrity Attacks", IEEE Transactions on Automatic Control, 2015.
> >
> > [6] Giraldo et al., "A Survey of Physics-Based Attack Detection in Cyber-Physical Systems", ACM Computing Surveys, 2018.

---

> ### Author Response · Authors · 2025-11-27
> **Kind Request for Feedback**
>
> Dear Reviewer i3Yb,
>
> As the discussion period will end in a few days, we would greatly appreciate it if you could review our responses and let us know if there are any further concerns or issues that we can address. Please also take a look at our revised draft, which improves the paper based on your comments and includes additional real-world power-system experiments on pages 27–30. We hope that you will kindly reconsider your score if our responses are satisfactory. Many thanks.
>
> Best,
>
> Authors

---

### Meta-Review · Area_Chair_hcPE · 2025-12-09

**Summary:**

This paper studies the identification of the input-output mapping of general nonlinear dynamical systems under adversarial attacks. They significantly broaden the scope of admissible control inputs and allow correlated, nonzero-mean, adversarial disturbances. Some theories are provided in this paper.

**Reviewer Concerns:**

Assumption 2.8 sticks out like a sore thumb.

Function approximation is a big issue.

Motivation is rather weak.

**Reviewer Scores:**

Reviewer i3Yb and Reviewer 7vZm may not change the score.

 The answers to Assumption 2.8  and motivation are not very well.

---

### Decision · Program_Chairs · 2026-01-26

Reject